# Small-scale structure of thermodynamic phase in Arctic mixed-phase clouds observed by airborne remote sensing during a cold air outbreak and a warm air advection event

Elena Ruiz-Donoso[1], André Ehrlich[1], Michael Schäfer[1], Evelyn Jäkel[1], Vera Schemann[2], Susanne Crewell[2], Mario Mech[2], Birte Solveig Kulla[2], Leif-Leonard Kliesch[2], Roland Neuber[3], and Manfred Wendisch[1]

[1]Leipzig Institute for Meteorology (LIM), University of Leipzig, Germany
[2]Institute for Geophysics and Meteorology, University of Cologne, Germany
[3]Alfred-Wegener-Institue Helmholtz Center for Polar and Marine Research (AWI), Germany

**Correspondence:** Elena Ruiz-Donoso (elena.ruiz_donoso@uni-leipzig.de)

**Abstract.** The combination of downward-looking airborne lidar, radar, microwave, and imaging spectrometer measurements was exploited to characterize the vertical and small-scale (down to 10 m) horizontal distribution of the thermodynamic phase of low-level Arctic mixed-layer clouds. Two cloud cases observed in a cold air outbreak and a warm air advection event observed during the Arctic CLoud Observations Using airborne measurements during polar Day (ACLOUD) campaign were
investigated. Both cloud cases exhibited the typical vertical mixed-phase structure with mostly liquid water droplets at cloud top and ice crystals in lower layers. The horizontal, small-scale distribution of the thermodynamic phase as observed during the cold air outbreak is dominated by the liquid water close to the cloud top and shows no indication of ice in lower cloud layers. Contrastingly, the cloud top variability of the case observed during a warm air advection showed some ice in areas of low reflectivity or cloud holes. Radiative transfer simulations considering homogeneous mixtures of liquid water droplets
and ice crystals were able to reproduce the horizontal variability of this warm air advection. Large eddy simulations (LES) were performed to reconstruct the observed cloud properties, which were used subsequently as input for radiative transfer simulations. The LES simulations of the cloud case observed during the cold air outbreak, with mostly liquid water at cloud top, realistically reproduced the observations. For the warm air advection case, the simulated ice water content ($IWC$) was systematically lower than the measured $IWC$. Nevertheless, the LES simulations revealed the presence of ice particles close to the cloud top and confirmed the observed horizontal variability of the cloud field. It is concluded that the cloud top small-
scale horizontal variability is directly linked to changes in the vertical distribution of the cloud thermodynamic phase. Passive satellite-borne imaging spectrometer observations with pixel sizes larger than 100 m miss the small-scale cloud top structures.

## 1   Introduction

In the Arctic, low-level stratus and stratocumulus clouds are present around 40 % of the time on annual average (Shupe et al.,
2006; Shupe, 2011) and they may persist up to several weeks (Shupe, 2011; Morrison et al., 2012). At least 30 % of these clouds are of mixed-phase type (Mioche et al., 2015). Their radiative properties and life cycles are determined by the partitioning and

the spatial (vertical/horizontal) distribution of liquid water droplets and ice crystals. Therefore, mixed-phase cloud properties are important for the characteristics of the Arctic climate system (Tan and Storelvmo, 2019). They are suspected to play an important role in the accelerated warming relative to lower latitudes observed in the last decades, a phenomenon known as Arctic amplification (Serreze and Barry, 2011; Wendisch et al., 2017). The microphysical and optical properties of Arctic

mixed-phase clouds are determined by a complex network of feedback mechanisms between local and large-scale dynamical and microphysical processes (e.g., Morrison et al., 2012; Mioche et al., 2017). Large-scale advection of air masses across the Arctic predefine their general nature (Pithan et al., 2018). In case of cold air masses advected from the central Arctic region towards lower latitudes, the cold air transported over the warm ocean surface produces intense shallow convection and characteristic cloud street structures, which may extend over several hundred kilometers. Cold air outbreaks occur all year long,

but they are especially frequent in winter (Kolstad et al., 2009; Fletcher et al., 2016). Warm and moist air masses intruding into the Arctic from southern latitudes occur 10 % of the time all year long and are responsible for most of the transport of moisture and heat into the Arctic (Woods et al., 2013; Sedlar and Tjernström, 2017; Pithan et al., 2018). During the northward transport, important air mass transformations take place. The air rapidly cools close to the surface, leading to shallow but strong temperature inversions promoting low-level, persistent clouds (Sedlar and Tjernström, 2017; Tjernström et al., 2015). In

these clouds, the vertical motion is driven mainly by radiative cooling at cloud top. As a consequence, convective cells appear in intervals of several kilometers (Shupe et al., 2008; Roesler et al., 2017). On smaller scales of a few hundred meters, the vertical motion is additionally driven by evaporative cooling, associated with entrainment of moist air supplied from upper layers (Mellado, 2017). This entrainment process ensures the formation of liquid water droplets and balances the loss of cloud water by precipitating ice crystals (Korolev, 2007; Shupe et al., 2008; Morrison et al., 2012). Observations by Schäfer et al.

(2017, 2018) show that the small-scale horizontal inhomogeneities of updrafts and downdrafts have typical length scales down to 60 m. In downdraft regions, the Wegener-Bergeron-Findeisen process may dominate over the nucleation of liquid water droplets (Korolev and Field, 2008; Korolev et al., 2017), causing the ice crystals to grow at the expense of the liquid water droplets.

Interactions between these processes determine the structure of the cloud, both vertically and horizontally. The cloud ther-

modynamic phase develops vertically in specific patterns. Most frequently, a liquid-water-dominated layer is observed from which ice crystals precipitate (Shupe et al., 2006; McFarquhar et al., 2007; Ehrlich et al., 2009; Mioche et al., 2015). Spatial differences of the cloud phase vertical distribution can, in turn, occur on horizontal scales down to tens of meters (Korolev and Isaac, 2006; Lawson et al., 2010). Therefore, understanding the radiative properties and temporal evolution of Arctic mixed-phase clouds requires a three-dimensional (3D) characterization of the thermodynamic phase partitioning, which relates the

vertical distribution of liquid droplets and ice crystals to the small scale structures observed close to the cloud top.

The analysis of small-scale microphysical inhomogeneities of Arctic stratus is challenging. Global climate models (GCMs) typically have horizontal and vertical grid sizes of 100 km and 1 km, respectively (Tan and Storelvmo, 2016). Global reanalysis products are provided with a horizontal grid that is typically larger than 40 km (Lindsay et al., 2014). This coarse resolution cannot resolve in-cloud microphysical and dynamical processes, such as the updraft and downdraft motions. Therefore, these

processes need to be parameterized (Field et al., 2004; Klein et al., 2009). Cloud resolving models (1 km horizontal and 30 m

vertical resolution; Luo et al., 2008), and large eddy simulations (LES, below 100 m horizontal and 15 m vertical resolution; Loewe et al., 2017) resolve small-scale cloud processes and are used to improve the GCMs subgrid mixed-phase cloud parameterization. In order to evaluate the performance of these high resolution simulations, adequately resolved observations are needed (Werner et al., 2014; Roesler et al., 2017; Schäfer et al., 2018; Egerer et al., 2019; Neggers et al., 2019; Schemann and
Ebell, 2020).

In the past, the observation of the thermodynamic phase transitions associated with small-scale cloud structures down to scales of 10 m was challenging due to limitations of the measurement methods. Passive and active satellite-borne remote sensing techniques have typical resolutions coarser than 250 m (Stephens et al., 2002). Ground-based active cloud remote sensing methods (lidar and radar), with vertical resolution of about 50 m and averaging intervals of 10 s (Kollias et al., 2007; Maahn
et al., 2015), mostly point only in zenith direction and thus may miss horizontal inhomogeneities (Marchand et al., 2007). Similarly, airborne in situ measurements of cloud microphysical properties require averaging periods of at least 1 s, integrating over scales of 50 m at a typical flight speed of $50\,\mathrm{m\,s^{-1}}$ (Mioche et al., 2017), and therefore, potentially mix individual pockets of ice crystals and liquid water droplets. Airborne active radar and lidar measurements also average over along-track distances of about 50 m (1 s at $50\,\mathrm{m\,s^{-1}}$ flight speed; Stachlewska et al., 2010; Mech et al., 2019). Airborne imaging remote sensing tech-
niques have the potential to map the cloud top geometry in high spatial resolution. Solar radiation measurements by spectral imagers provide data with an spatial resolution of down to a few meters. Based on this measurement approach, Schäfer et al. (2013) and Bierwirth et al. (2013) retrieved two-dimensional (2D) fields of cloud optical thickness resolving changes in spatial scales smaller than 50 m, which are associated with the evaporation of cloud particles in downdraft regions. For selected cases, Thompson et al. (2016) illustrated the potential of spectral imagers to retrieve 2D fields of cloud thermodynamic phase. The
identification of mixed-phase cloud regions, however, was based on the assumption of homogeneously mixed clouds and did not consider the vertical distributions of the ice crystals and liquid water droplets. Due to the passive nature of the imaging spectrometers, the measurements integrate over the entire cloud column, although they are dominated by the cloud properties close to the cloud top (Platnick, 2000). They commonly cannot resolve the clouds vertically. Therefore, to avoid misclassifications, the information about the cloud vertical structure provided by active remote sensing is needed to interpret passive remote
sensing measurements of reflected solar radiation.

This study exploits combined passive spectral imaging techniques and active remote sensing measurements (radar and lidar) to characterize the cloud phase partitioning in the 3D cloud structure. The active remote sensing instruments provide the general vertical stratification of ice particles and liquid water droplets, which is needed to interpret the 2D maps of cloud phase observed by the spectral imager. Two mixed-phase cloud cases detected during the Artic CLoud Observations Using
airborne measurements during polar Day (ACLOUD) campaign are chosen to demonstrate this instrument synergy (Wendisch et al., 2019). Section 2 introduces the instrumentation, the retrieval approach to derive 2D maps of cloud phase, and the LES simulations. The two case studies are presented in Section 3, including a discussion of the impact of the cloud vertical structure on the cloud phase retrieval. The observation are compared to LES simulations in Section 4. The information loss due to the smoothing of the fine-scale cloud structures to the typical geometry obtained by satellite-borne remote sensing is quantified in
Section 5.

## 2 Methods

### 2.1 Observations

The ACLOUD campaign was performed to improve the understanding of the role of Arctic low and mid-level clouds in Arctic amplification; it took place in the vicinity of the Svalbard archipelago in May and June 2017 (Wendisch et al., 2019; Ehrlich et al., 2019). During ACLOUD, active and passive remote sensing instruments and in-situ probes were operated on the research aircraft Polar 5 and Polar 6 of the Alfred Wegener Institute Helmholtz-Center for Polar and Marine Research (AWI; Wesche et al., 2016). Among the in-situ probes installed on Polar 6, the Small Ice Detector (SID-3, Vochezer et al., 2016) provides the particle size distribution of hydrometeors with sizes between $5\,\mu$m and $45\,\mu$m. The passive remote sensing equipment installed on Polar 5 included, among others, the AISA Hawk spectral imager (Pu, 2017). The downward-viewing pushbroom sensor of AISA Hawk is aligned across-track to measure 2D fields of upward radiance ($I_\lambda^\uparrow$) reflected by the cloud and surface. Considering uncertainties due to the calibration and noise in the measured signal, the uncertainty in the measured radiance is estimated to be in the range of 6 % (Schäfer et al., 2013). With 384 across-track pixels, a $36°$ field of view (FOV) and a typical vertical distance between aircraft and cloud top of $1\,$km, AISA Hawk samples with a spatial resolution of roughly $2\,$m. At this resolution, horizontal photon transport needs to be taken into account. The AISA Hawk measurements have been corrected from this effect using the deconvolution algorithm introduced in App. A. Each pixel contains spectral measurements between $930\,$nm and $2550\,$nm wavelength in 288 channels with an average spectral resolution (full width at half maximum, FWHM) of about $10\,$nm. More details on the calibration of AISA Hawk and the data processing are presented by Ehrlich et al. (2019). Two-dimensional fields of spectral cloud top reflectivity ($R_\lambda$) are obtained by combining reflected radiance fields, detected by AISA Hawk, with simultaneous measurements of the downward spectral irradiance ($F_\lambda^\downarrow$) obtained by the Spectral Modular Airborne Radiation measurement sysTem (SMART; Wendisch et al., 2001; Ehrlich et al., 2019):

$$R_\lambda = \pi \cdot \frac{I_\lambda^\uparrow}{F_\lambda^\downarrow}. \tag{1}$$

The cloud top reflectivity $R_\lambda$ in the spectral range between $\lambda_\mathrm{a} = 1550\,$nm and $\lambda_\mathrm{b} = 1700\,$nm, characterized by the different absorption features of liquid water and ice, is used to discriminate the cloud thermodynamic phase (Pilewskie and Twomey, 1987; Chylek and Borel, 2004; Jäkel et al., 2013; Thompson et al., 2016). The spectral differences in the cloud top reflectivity of pure liquid and pure ice clouds are illustrated in Fig. 1. To identify the cloud phase, Ehrlich et al. (2008a) defined the slope phase index ($\mathcal{I}_\mathrm{s}$), which quantifies the spectral slope of the cloud top reflectivity in this spectral region and is sensitive to the amount of ice crystals and liquid water droplets close to cloud top:

$$\mathcal{I}_\mathrm{s} = 100 \cdot \frac{(\lambda_\mathrm{b} - \lambda_\mathrm{a})}{R_{1640}} \left( \frac{\mathrm{d}R_\lambda}{\mathrm{d}\lambda} \right)_{[\lambda_\mathrm{a}, \lambda_\mathrm{b}]}. \tag{2}$$

A threshold value for the slope phase index of 20 discriminates between pure liquid water ($\mathcal{I}_\mathrm{s} < 20$) and pure ice or mixed-phase ($\mathcal{I}_\mathrm{s} > 20$) close to cloud top (Ehrlich et al., 2009). By applying Eq. (2) to the AISA Hawk measurements, fields of $\mathcal{I}_\mathrm{s}$ are generated, which resolve the horizontal distribution of the thermodynamic phase of the cloud uppermost $200\,$m layer, typically corresponding to an in-cloud optical depth of about 5 (Platnick, 2000; Ehrlich, 2009; Miller et al., 2014).

The vertical distribution of the cloud thermodynamic phase is retrieved from measurements by the Microwave Radar/radiometer for Arctic Clouds (MiRAC; Mech et al., 2019) and the Airborne Mobile Aerosol Lidar (AMALi; Stachlewska et al., 2010) deployed in parallel with the AISA Hawk sensor on board of Polar 5. The radar reflectivity is proportional to the sixth power of the particle size distribution, and thus, is most sensitive to large particles, such as ice crystals (Hogan and O'Conner, 2004; Shupe, 2007; Kalesse et al., 2016). Therefore, it is used as an indicator of the vertical location of large ice crystals in mixed-phase clouds. In contrast, the AMALi backscatter signal is strongly attenuated by high concentrations of small particles and, thus, identifies the location of small supercooled liquid water droplets close to the cloud top in mixed-phase clouds.

## 2.2 Radiative transfer modelling

Radiative transfer simulations are employed to interpret the horizontal structure of the slope phase index, and to retrieve 2D fields of cloud optical thickness ($\tau$) and effective radius ($r_{\mathrm{eff}}$). They were performed with the Library for Radiative transfer (libRadtran) code (Mayer and Kylling, 2005; Emde et al., 2016). The simulations applied the radiative transfer solver FDISORT2 (Discrete Ordinate Radiative Transfer) introduced by Stamnes et al. (2000). The standard sub-Arctic summer atmospheric profile provided by libRadtran was employed, together with temperature and water vapor profiles measured by dropsondes released during the respective flights close to the measurement sites. A maritime aerosol type and the surface albedo of open ocean were selected (Shettle, 1990). The solar zenith angle (SZA) was adjusted to the location and time of each specific measurement. The simulations of liquid water clouds assumed the validity of Mie theory, whereas those including ice clouds assumed columnar ice crystals and applied the "Hey" parameterization, based on Yang et al. (2000) to convert microphysical into optical properties. Regarding the phase index, Ehrlich et al. (2008a, b) found that the influence of the ice crystal shape is of minor importance compared to the impact of the particle size, which was confirmed by additional simulations considering different ice crystal habits (not shown here). Hence, the assumption of columns is sufficient to account for the non-sphericity effects of the ice crystals.

In a first step, extending the work of Bierwirth et al. (2013) and Schäfer et al. (2013) to the near infrared spectral range, the spectral cloud top reflectivity fields measured by AISA Hawk were used to retrieve fields of optical thickness and effective radius. For this purpose, the reflectivity $R_{1240}$ at a wavelength of 1240 nm (scattering dominated), sensitive to the cloud optical thickness, is combined with $R_{1625}$ at a wavelength of 1625 nm, where absorption of solar radiation dominates and influenced mainly by the particle size (Nakajima and King, 1990). The location of these wavelengths in the cloud top reflectivity spectrum is shown in Fig. 1. To reduce the retrieval uncertainties, the radiance ratio approach by Werner et al. (2013) was applied. Look-up tables considering the sensor viewing geometry of every pixel of AISA Hawk are simulated for various combinations of cloud optical thickness and effective radius. For the simulations, pure liquid water clouds are assumed. Therefore, in the case of mixed-phase clouds, the retrieved values of optical thickness and effective radius might be biased. However, since Arctic low-level mixed-phase clouds are typically topped by a liquid-water layer (Shupe et al., 2006; McFarquhar et al., 2007), the associated uncertainties are expected to be lower than the variability within the cloud field.

The retrieved optical thickness and effective radius, assuming a plane-parallel 1D radiative transfer model, are affected by 3D radiative effects (Zinner and Mayer, 2006; Marshak et al., 2006). While the 3D nature of the cloud structures will cause an

overestimation of the optical thickness in the brightly illuminated areas, the effective radius is overestimated in the shadowed regions. Horváth et al. (2014) showed that, due to their opposite sign, the 3D bias of retrieved optical thickness and effective radius partially cancel when calculating the liquid water path *LWP*. Therefore, the retrieved fields of $\tau$ and $r_{\mathrm{eff}}$ are converted into fields of *LWP* using the relation by Kokhanovsky (2004):

$$LWP = \frac{2}{3} \cdot \rho \cdot \tau \cdot r_{\mathrm{eff}}. \tag{3}$$

As it was the case for the retrieved $\tau$ and $r_{\mathrm{eff}}$, this conversion assumes liquid water clouds with a homogeneously mixed vertically constant profile. Considering a homogeneous vertical profile may result in inaccuracies even for pure liquid water clouds (Zhou et al., 2016). Mixed-phase clouds, in addition, violate the pure-phase assumption. The presence of ice crystals introduces a significant error in the calculated $LWP$, which reaches values well above the typical values observed in Arctic pure liquid water clouds. Past observations show that the $LWP$ of typical Arctic boundary-layer clouds is in the range of 30 - 50 $\mathrm{g\,m^{-2}}$ and rarely exceeds $100\,\mathrm{g\,m^{-2}}$, (Shupe et al., 2006; Mioche et al., 2017; Nomokonova et al., 2019; Gierens et al., 2019). Appendix A analyzes the different impact of shades and inhomogeneous thermodynamic phase distributions in the retrieved $LWP$. In this paper, unrealistically high retrieved *LWP* values are used to identify mixed-phase clouds.

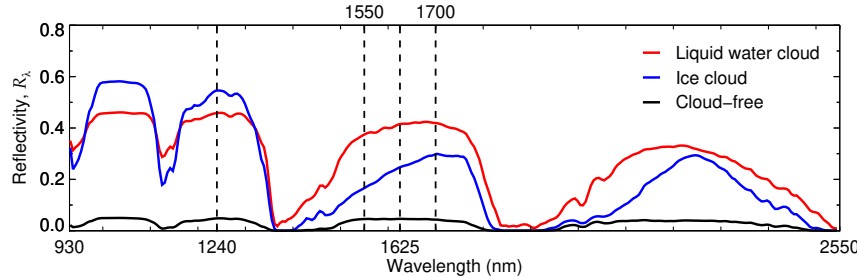

**Figure 1.** Reflectivity spectra of a pure liquid water cloud and a pure ice cloud of optical thickness 12 compared with a clear sky spectrum in the wavelength range measured by AISA Hawk. The vertical dashed lines indicate the wavelengths needed to calculate the slope phase index (1550 - 1700 nm), and to retrieve the cloud optical thickness (1240 nm) and effective radius (1625 nm)

### 2.3 Large Eddy Simulations (LES)

Simulations using the ICOsahedral Non-hydrostatic atmosphere model (ICON), operated in its Large Eddy Model (LEM) configuration (Heinze et al., 2017; Dipankar et al., 2015), provide a quantitative view into the cloud vertical structure. The simulated cloud vertical profiles were used as input for radiative transfer simulations to analyze the impact of different vertical distributions of the cloud thermodynamic phase on the cloud top horizontal variability.

ICON-LEM simulations were forced by initial and lateral boundary conditions from the European Centre for Medium-Range Weather Forecasts (ECMWF) Integrated Forecast System (IFS; Gregory et al., 2010). The simulations were preformed in a one-way nested setup with a 600 m spatial resolution at the outermost domain, followed by 300 m resolution and an inner

triangular nest of 150 m resolution. This inner nest was equivalent to a square grid of 100 m horizontal resolution, which is about one order of magnitude coarser than the observations by AISA Hawk. Simulations with finer horizontal resolution were not reasonable due to the high computational time. In the vertical direction, 150 height levels were simulated. In the ICON-LEM simulations the two-moment mixed-phase bulk microphysical parameterization by Seifert and Beheng (2006) was applied. It

provided vertical profiles of liquid and ice mass mixing ratio $r_w$, $r_i$, cloud droplets and ice crystal number concentration $N_w$, $N_i$, air temperature $T$, and pressure $p$. The mass mixing ratio and the number concentration profiles take into consideration both, the non-precipitating (cloud water and cloud ice) and the precipitating (rain, snow, graupel and hail) hydrometeors. They have been used to convert the $r_w$ and $r_i$ into $LWC$, and $IWC$, as required by the radiative transfer model:

$$LWC(z) = r_w(z) \cdot \frac{p(z)}{R \cdot T(z)}, \quad IWC(z) = r_i(z) \cdot \frac{p(z)}{R \cdot T(z)} \tag{4}$$

with R = 287.06 J kg$^{-1}$ K$^{-1}$ the specific gas constant for dry air, and $z$ the altitude. For the spherical liquid water droplets, vertical profiles of droplet effective radius are obtained by (Martin et al., 1994; Kostka et al., 2014):

$$r_{\text{eff,liquid}}(z) = \left[ \frac{3 \cdot r_w(z)}{4 \cdot \pi \cdot \rho_w \cdot N_w(z)} \right]^{1/3}, \tag{5}$$

where $\rho_w$ is the density of the liquid water. For the non-spherical ice crystals, the median mass diameter $D_{\text{m,ice}}$ of the particle size distribution (PSD) of cloud ice represented by the generalized $\Gamma$-distribution described by Seifert and Beheng (2006), used

by ICON-LEM, is calculated as:

$$D_{\text{m,ice}}(z) = a \cdot \left[ \frac{r_i(z)}{N_i(z)} \right]^b, \tag{6}$$

with a = 0.206·10$^{-6}$ m kg$^{-b}$ and b = 0.302. The radiative properties of ice crystals were parameterized using the effective radius $r_{\text{eff,ice}}$. To convert the median particle size into radius $r_{\text{eff,ice}}$, the measurement-based relationship between $D_{\text{m,ice}}$ and the effective diameter, $D_{\text{eff,ice}}$, of columnar ice crystals introduced by Baum et al. (2005) and Baum et al. (2014) was used.

## 3    Results of measurements and radiative transfer simulations

The ACLOUD campaign was classified by Knudsen et al. (2018) into a cold (May 23 - May 29), a warm (May 30 - June 12) period, and a neutral period (June 13 - June 26). During the cold period, the Svalbard region was affected by a northerly cold air outbreak, which led to the development of low-level clouds over the warm open ocean. Over the Fram Strait, these clouds organized in a roll convective structure, forming typical cloud streets. During the warm period, a high pressure system

south of Svalbard advected warm air from the south over the archipelago, leading to the development of a low-level, optically thick and homogeneous stratocumulus. Cold air outbreaks and warm air advections are phenomena often affecting the Arctic regions (Pithan et al., 2018; Sedlar and Tjernström, 2017; Woods et al., 2013; Kolstad et al., 2009; Fletcher et al., 2016). The occurrence of both situations during the ACLOUD campaign make it an ideal testbed to contrast the characteristics of the clouds occurring under each situation. Two cloud cases observed on 25 May, during the cold air outbreak, and 2 June 2017,

during the warm air advection, were analyzed in detail. Figure 2 displays the corresponding MODerate resolution Imaging Spectroradiometer (MODIS) true color images showing the clouds on both days.

Figure 3 illustrates the combined measurements of MiRAC and AMALi for the one-minute sequence acquired over open ocean for the two cloud cases. The combination of measurements is interpreted qualitatively to gain an insight into the clouds
vertical structure. In both cases, the liquid cloud top is well identified by the strong backscatter of the lidar signal, defined as in Langenbach et al. (2019) and highly sensitive to liquid droplets. Whereas on 25 May the liquid layer is geometrically thicker, the lidar reaches the surface, which indicates a cloud optical thickness less than 3-4 (McGill et al., 2004). On 2 June, the lidar cannot penetrate the cloud. The stronger attenuation of the lidar signal, i.e., the rapid decrease in the lidar backscatter, hints at larger amounts of liquid than on 25 May. In contrast, the radar signal is dominated by larger particles, and higher radar
reflectivity values commonly indicate higher concentrations of ice crystals. The combination of the radar and lidar signals helps to identify differences in the vertical structure of both clouds. The cloud on 25 May, showing a high radar reflectivity, contains very likely precipitating large ice crystals. In this case, some regions of the cloud are characterized by a large radar reflectivity at cloud top, shown by the overlapping radar and lidar signals in Fig. 3a, which hints at the presence of large particles in high cloud layers. Vertical separation between the signals of both instruments, such as occurring around 9:01:47, indicate regions
where small liquid droplets dominate the cloud top, detected by the lidar but not by the radar. In these regions, the radar observes large particles, likely ice crystals, around 100 m below the cloud top which precipitate down to the surface. On 2 June (Fig. 3b), the radar reflectivity is weaker than on 25 May and shows no evidence of precipitation reaching the surface. The weaker radar reflectivity may be attributed either to smaller ice crystals or to a reduced particle concentration. However, the continuous overlap between the lidar and the radar signals in Fig. 3 indicates the presence of large particles right below the
cloud top. These differences in the vertical structures of the two cloud cases need to be considered when interpreting the 2D horizontal fields of the slope phase index retrieved by AISA Hawk, which is most sensitive to the cloud top layer.

## 3.1  Cold air outbreak

Figure 4 presents a sequence of AISA Hawk measurements and retrieved horizontal fields of cloud properties ($R_{1240}$, $\mathcal{I}_\mathrm{s}$, $\tau$, $r_\mathrm{eff}$, and $LWP$) together with corresponding histograms. They were observed during the cold air outbreak on 25 May 2017 in
the flight section shown in Figs. 2a and 2b, simultaneously with the MiRAC and AMALI observations in Fig. 3a. Mean values and associated uncertainty of the cloud properties are summarized in Tab. 1. The measurements present one minute of data acquired at 9:01 UTC with a SZA of 60.5° at a flight altitude of 2.8 km. The average cloud top was located at 400 m above sea level. The observed cloud scene covers an area of $1.1 \times 4.7\,\mathrm{km}^2$ with an average pixel size of $3.9 \times 2.6\,\mathrm{m}^2$. Figure 4a shows the cloud top reflectivity field at 1240 nm wavelength, $R_{1240}$, and a corresponding histogram in Fig. 4b. Due to the broken
character of the cloud field, a cloud mask has been applied prior to the retrieval of cloud properties. Based on radiative transfer simulations, a threshold of $R_{1240} = 0.1$, roughly corresponding to a $LWP$ of $2\,\mathrm{g\,m}^{-2}$, was chosen to discriminate between cloudy and cloud-free areas. Regions with $R_{1240} < 0.1$ were classified as cloud-free and have been excluded from further analysis.

The slope phase index $\mathcal{I}_\mathrm{s}$, presented in Figs. 4c and 4d, shows a maximum value of 12.6, which is characteristic for pure liquid water clouds. This seems to disagree with the lidar and radar observations (Fig. 3), which indicated a mixed-phase cloud,

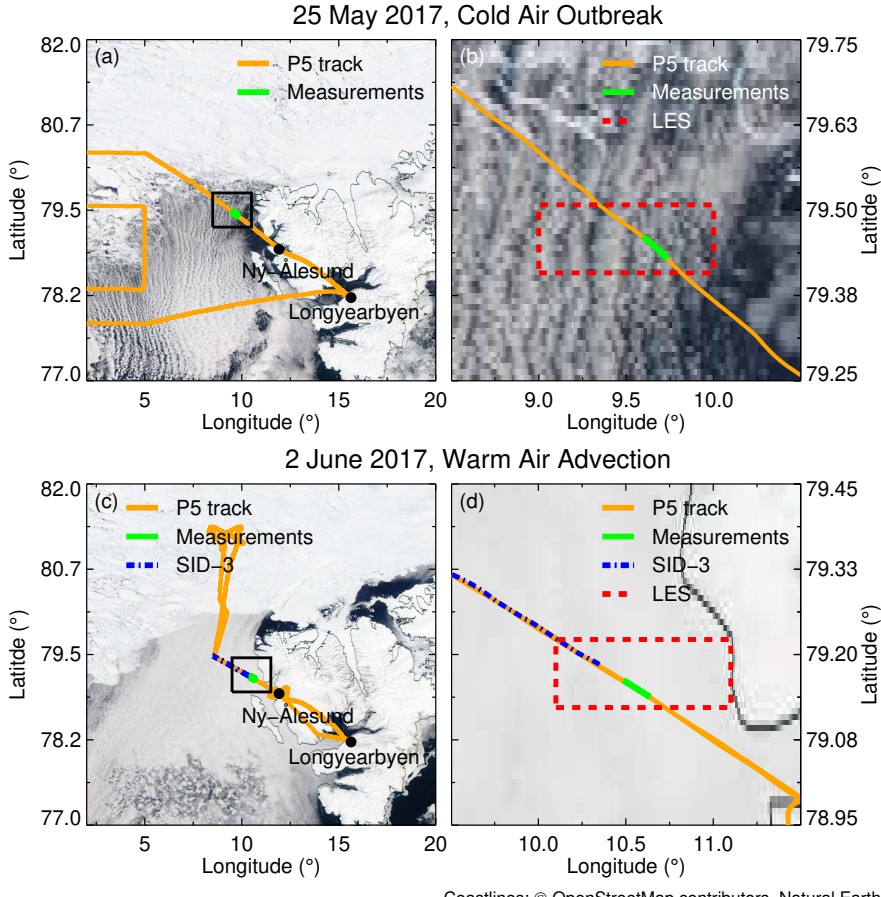

**Figure 2.** MODIS true color images from the NASA Worldview application (https://worldview.earthdata.nasa.gov) on 25 May 2017 (**a**) during a cold air outbreak and on 2 June 2017 (**c**) during a warm air advection. Zooms into the regions delimited by black squares are shown in (**b**) and (**c**). The measurements location (79.5° N, 9.5° E on 25 May and 79.2° N, 10.7° E on 2 June) is indicated by the green section of the flight track of Polar 5 (orange). The areas extracted from the LES are indicated by the dashed red rectangle. The dashed-dotted blue on 2 June line indicates the location of the SID-3 measurements.

and demonstrates the higher sensitivity of the phase index to the thermodynamic phase of the top most layer. Similarly, the *LWP* (Fig. 4i), calculated from $\tau$ (Fig. 4e) and $r_{\mathrm{eff}}$ (Fig. 4g) using Eq. (3), increases towards the cloud core centers, as it is typical for pure liquid water clouds. These areas visually identify updraft regions where enhanced condensation occurs due to adiabatic cooling (Gerber et al., 2005).

5     Although $\mathcal{I}_s$ is always below the threshold of pure ice clouds, the cloud field presents significant small-scale variability that might be related to spatial changes in the thermodynamic phase distribution. To quantify if regions of enhanced $\mathcal{I}_s$ are correlated with areas of precipitating ice crystals, as observed by MiRAC, the cloud edges were separated from the central

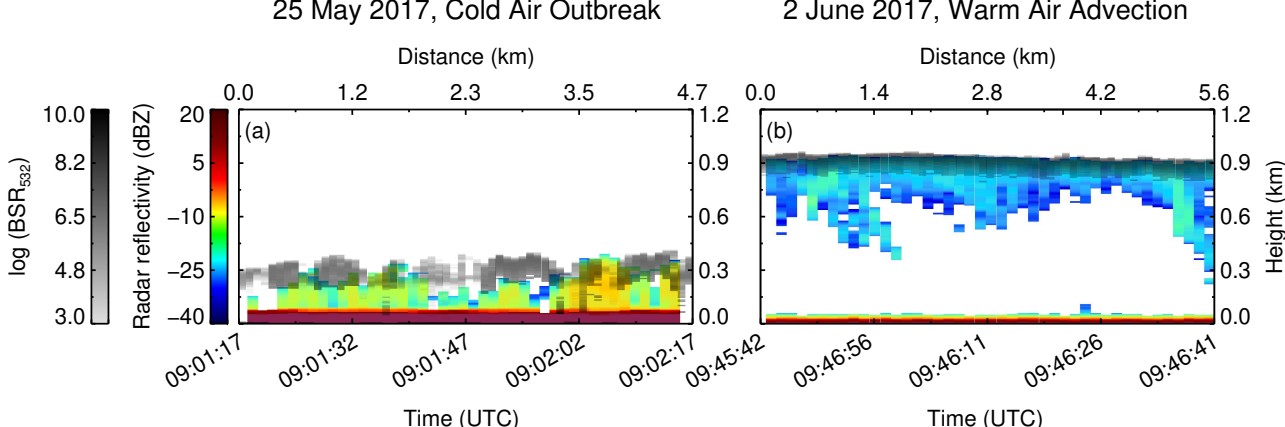

**Figure 3.** Combination of MiRAC radar reflectivity (color range between blue and red) and AMALi backscatter ratio (colors between white and black) as measured on 25 May 2017 during a cold air outbreak (**a**) and on 2 June 2017 during a warm air advection (**b**). AMALi's lidar backscatter ratio is highly sensitive to the liquid droplets and shows the liquid top layer in both clouds. MiRAC's radar reflectivity is dominated by larger particles and indicate regions with ice crystals. The radar signal below an altitude of 150 m is heavily influenced by ground clutter and cannot interpreted for cloud studies.

cloud regions. All pixels below the 25th percentile of $R_{1240}$ and of $\mathcal{I}_\mathrm{s}$ are defined as cloud edges. All other areas are considered to be cloud core center regions. The separated measurements were compared to 1D radiative transfer simulations adapted to the measurement situation. In Fig. 5, the measured slope phase index is presented as a function of the cloud top reflectivity, together with simulations assuming pure-phase (either liquid or ice) clouds of known particle sizes and liquid/ice water paths.

5    This sensitivity study shows the spread of $\mathcal{I}_\mathrm{s}$ as a function of the cloud thermodynamic phase, the cloud optical thickness (or *LWP*, *IWP*), and the cloud particle size. An accurate phase classification cannot rely on a fixed $\mathcal{I}_\mathrm{s}$ threshold value and depends on the combined $\mathcal{I}_\mathrm{s}$ and $R_{1240}$ values. Fig. 5 reveals that the observed $\mathcal{I}_\mathrm{s}$ and $R_{1240}$ range within simulated values covered by pure liquid water clouds. The spatio-temporal changes of the measurement (color code in Fig. 5) indicate that a transition from cloud edge into cloud core follows lines with increasing *LWP* and slightly increasing particle sizes. This pattern can be

10    explained by the dynamical and microphysical processes in cloud cores where ascending air condenses and cloud droplets grow with altitude leading to a higher *LWP*. Hence, the small-scale variability of $\mathcal{I}_\mathrm{s}$ observed on 25 May 2017 can be interpreted as the natural variability of the cloud top liquid layer. Compared to the radar observations, the passive reflectivity measurements are insensitive to the precipitating ice crystals.

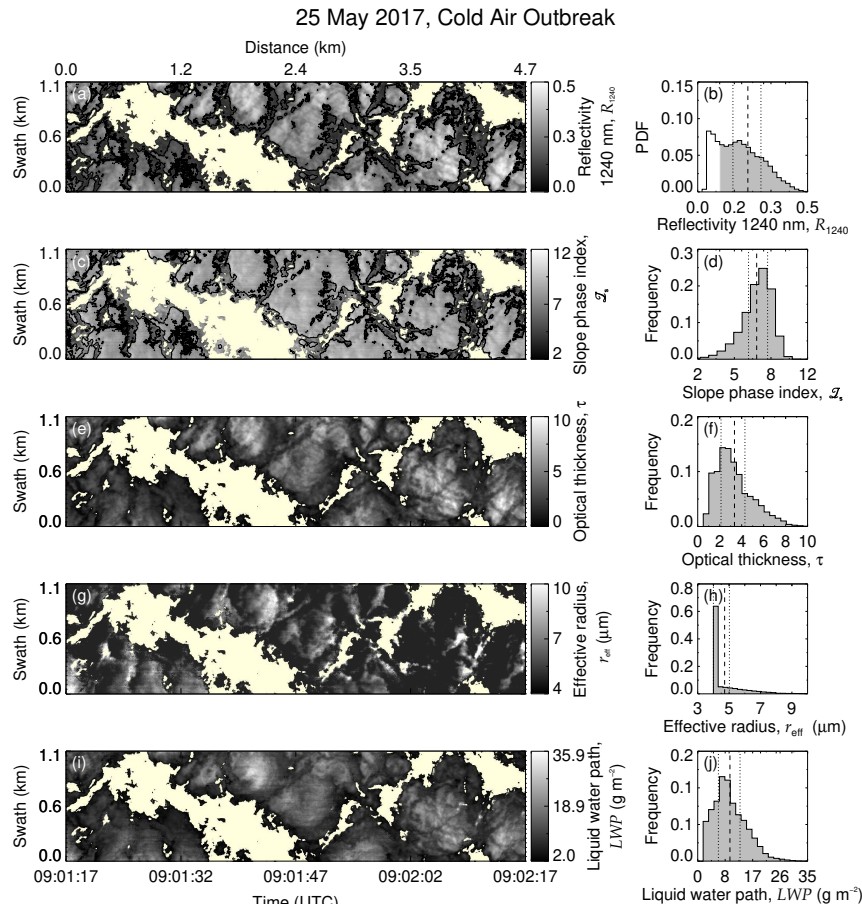

**Figure 4.** AISA Hawk measurement on 25 May 2017. Cloud top reflectivity (**a**), slope phase index (**c**), retrieved optical thickness (**e**), retrieved effective radius (**g**) and liquid water path (**i**). The overlayed contours in (**a**) and (**c**) separate the cloud central regions from the cloud edges. The frequency of occurrence histograms are displayed on the corresponding right-hand figures (**b, d, f, h, j**). Data classified as cloud free is shown by the non-colored histogram in (**b**). Dashed lines indicates the mean value of each field and the dotted lines show the corresponding 25th and 75th percentile.

## 3.2 Warm air advection

### 3.2.1 2D horizontal fields

A sequence of $R_{1240}$ and retrieved cloud properties ($\mathcal{I}_s$, $\tau$, $r_{eff}$, *LWP*) observed in the ACLOUD warm period on 2 June 2017 is shown in Fig. 6 for the flight section of Fig. 3b. Table 1 presents the mean values and associated uncertainty of the presented cloud properties. The one-minute sequence starts at 9:45 UTC, when the SZA was of about 57.9°. The lidar observations indicated that the cloud top of the low-level stratocumulus was located at 900 m above sea level. Hence, for a

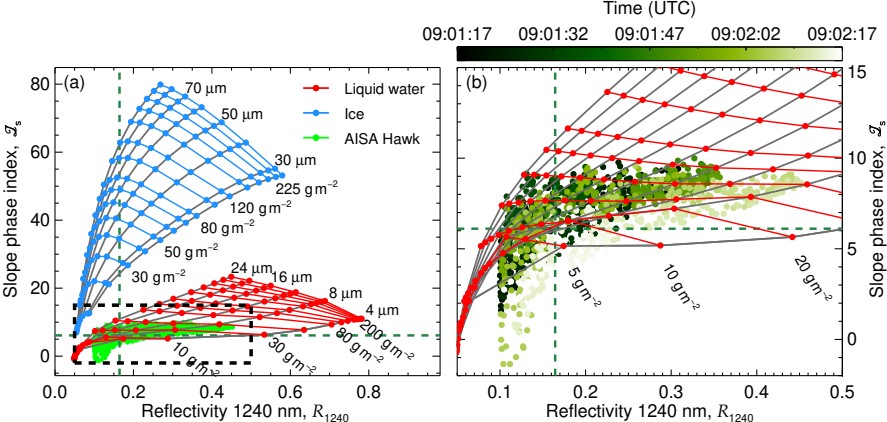

**Figure 5.** (a) $\mathcal{I}_\mathrm{s}$ measured on 25 May 2017 presented as a function of $R_{1240}$ (green dots). The dashed lines indicate the 25th percentile of $R_{1240}$ and $\mathcal{I}_\mathrm{s}$. The two grids represent radiative transfer simulations for a range of pure liquid (red) and pure ice (blue) clouds. The liquid water clouds cover droplets with $r_\mathrm{eff}$ between 4 and 24 $\mu$m and $LWP$ between 1 and 250 g m$^{-2}$. The ice clouds are simulated for columnar ice crystals with $r_\mathrm{eff}$ between 28 and 90 $\mu$m and IWP between 1 and 250 g m$^{-2}$. A SZA of 60.5° was considered. (b) Zoom of the area highlighted by a dashed rectangle in (a). Color-coded is the acquisition time of the measurements illustrating changes along the flight path.

**Table 1.** Average value and uncertainty ($\Delta$) of the cloud top properties derived from the measurements of AISA Hawk on 25 May and on 2 June. Independent estimations of the $LWP$ range by the passive 89 GHz channel of MiRAC are also included.

| | 25 May 2017 | 2 June 2017 |
|---|---|---|
| $z_\mathrm{top}$ (m) | 400 | 900 |
| SZA (°) | 60.5 | 57.9 |
| $\bar{R}_{1240} \pm \Delta\bar{R}_{1240}$ | $0.23 \pm 0.01$ | $0.65 \pm 0.03$ |
| $\bar{\mathcal{I}}_\mathrm{s} \pm \Delta\bar{\mathcal{I}}_\mathrm{s}$ | $7.36 \pm 0.04$ | $20.3 \pm 1.0$ |
| $\bar{\tau} \pm \Delta\bar{\tau}$ | $3.35 \pm 0.15$ | $33.7 \pm 4.8$ |
| $\bar{r}_\mathrm{eff} \pm \Delta\bar{r}_\mathrm{eff}\,(\mu\mathrm{m})$ | $4.7 \pm 1.5$ | $12.5 \pm 3.5$ |
| $\overline{LWP} \pm \Delta\overline{LWP}\,(\mathrm{g\,m}^{-2})$ | $10.3 \pm 3.7$ | $271 \pm 93$ |
| $LWP_\mathrm{MiRAC} \pm \Delta LWP_\mathrm{MiRAC}\,(\mathrm{g\,m}^{-2})$ | $20 \pm 1$ - $40 \pm 2$ | $90 \pm 5$ - $120 \pm 7$ |

flight altitude of 2.9 km, the field covers a cloud area of $0.9 \times 5.6\,\mathrm{km}^2$ with an average pixel size of $3.1 \times 4.7\,\mathrm{m}^2$. The cloud top reflectivity at 1240 nm wavelength, displayed in Fig. 6a, shows a rather horizontally uniform cloud layer compared to the measurements collected on 25 May 2017 (Case I). The cloud mask ($R_{1240} > 0.1$) reveals a 100 % cloud coverage for this scene. The slope phase index, presented in Fig. 6c, is higher compared to the cloud case presented in Fig. 4 and ranges between 14.9 and 36.5. Applying the common threshold of 20 would classify larger regions of the observed clouds as pure ice or mixed-phase. However, the $LWP$ (Fig. 6i) shows significant variability over the entire cloud field, which may be related to

the spatial distribution of the thermodynamic phase. The comparison of the relation between $\mathcal{I}_s$ and $R_{1240}$ with simulations assuming pure-phase clouds is shown in Fig. 7. The simulations reveal that the measurements do not fall in the range of the grid simulated for pure ice clouds, which would typically have higher values of slope phase index than observed. The measurements rather resemble the simulations of pure liquid water clouds. However, the field and histogram of $LWP$ (Figs. 6i and 6j) show values in the range of $270\,\mathrm{g\,m^{-2}}$ with 25 % percentile at $250\,\mathrm{g\,m^{-2}}$. Such high $LWP$ values have rarely been observed in Arctic low-level clouds, which typically ranges between 30 and $50\,\mathrm{g\,m^{-2}}$ and rarely exceed $100\,\mathrm{g\,m^{-2}}$ (Shupe et al., 2005; de Boer et al., 2009; Mioche et al., 2017; Nomokonova et al., 2019; Gierens et al., 2019). The measurements by the passive 89 GHz channel of the microwave radiometer of MiRAC were used to estimate the $LWP$ independently (see App. B for retrieval description and uncertainty assessment). The values between 90 and $120\,\mathrm{g\,m^{-2}}$ indicate that the $LWP$ retrieval using the AISA Hawk measurements is strongly overestimated likely due to the presence of ice crystals close to cloud top (compare Fig. 3). This is supported by the rather high optical thickness and particle sizes retrieved from AISA Hawk measurements, shown in Figs. 6e-h. As the retrieval assumes liquid droplets, the presence of ice crystals, which are typically larger and strongly absorb radiation at 1625 nm wavelength, bias the retrieval of both quantities towards higher values (Riedi et al., 2010). The particle size distribution observed by the SID-3 (Schnaiter and Järvinen, 2019) deployed in Polar 6 between 9:25 and 9:35 UTC in the vicinity of the AISA Hawk measurements (Fig. 2) revealed that, for the observed cloud, the particles at cloud top present effective radii in the range of $10\,\mu$m. 75 % of the AISA Hawk measurements on 2 June retrieve an effective radii larger than this value (Figs. 6g and 6h). The small-scale variability of the cloud properties shows that the largest deviation of the retrieved $r_{\mathrm{eff}}$ and $LWP$ respect the external measurements occurs in areas of low reflectivity (below the 25th percentile of $R_{1240}$) and high slope phase index values (above the 75th percentile of $\mathcal{I}_s$). These areas indicate cloud holes, where the vertical velocity is likely downwards and the condensation of liquid droplets is reduced, which increases the fraction of ice crystals. Although the theory predicts low values of $LWP$ and $r_{\mathrm{eff}}$ in these regions (Gerber et al., 2005, 2013), the high ice fraction leads to the strong overestimation of $LWP$ compared to the microwave retrieval. In contrast to the pattern observed on 25 May 2017, the higher ice fraction in the edges of the cloud holes causes the slope phase index to decrease with increasing cloud top reflectivity.

### 3.2.2 Impact of the vertical distribution of ice and water

Mixed-phase clouds in the Arctic commonly consist of a single layer of supercooled liquid water droplets at cloud top, from which ice crystals precipitate (Mioche et al., 2015), which is in line with the radar/lidar observations presented in Fig. 3. Additionally, Ehrlich et al. (2009) found evidence of ice crystals near the cloud top. Horizontal inhomogeneities in the vertical distribution of the liquid water and ice occur in horizontal scales of 10 m (Korolev and Isaac, 2006; Lawson et al., 2010) and are expected to relate to the small-scale structures (i.e., holes and domes) on the cloud top. Therefore, reproducing the observed trends of $R_{1240}$ and $\mathcal{I}_s$ with simulated mixed-phase clouds can provide information about the horizontal distribution of the cloud thermodynamic phase vertical structure. For this reason, the $R_{1240}$ and $\mathcal{I}_s$ observed on 2 June are compared with three different vertical mixing scenarios. A two-layer cloud scenario with a layer of liquid water droplets at cloud top (750 - 900 m) and a cloud bottom layer (600 - 750 m) consisting of precipitating ice particles was assumed to represent the common two-layers vertical thermodynamic phase distribution. In a second and third scenario, a vertically homogeneous mixture of ice

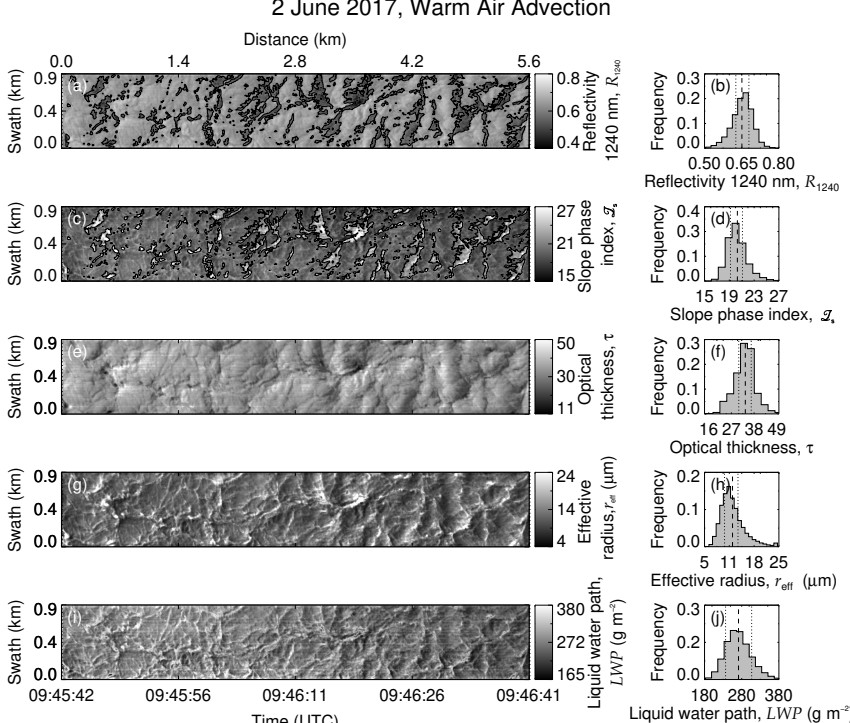

**Figure 6.** AISA Hawk measurement on 2 June 2017. Cloud top reflectivity (**a**), slope phase index (**c**), retrieved optical thickness (**e**), retrieved effective radius (**g**) and liquid water path (**i**). The overlayed contours in (**a**) and (**c**) separate the cloud central regions from the cloud edges. The frequency of occurrence histograms are displayed on the corresponding right-hand figures (**b, d, f, h, j**). The dashed line indicates the mean value and the dotted lines show its 25th and 75th percentile.

and liquid particles was assumed in the cloud layer (600 - 900 m), to represent the case when both liquid water and ice crystals are also present in the upper cloud top layer. The partitioning between ice and liquid droplets was varied by changing the ice fraction, defined by:

$$IF = \frac{IWP}{TWP} \cdot 100\,\%, \tag{7}$$

5  with the total water path defined as $TWP = LWP + IWP$. Pure liquid water clouds correspond to $IF = 0\,\%$ and pure ice clouds to $IF = 100\,\%$. The slope phase index and the spectral cloud top reflectivity depend on the $r_{\mathrm{eff}}$ of the ice and liquid particles and on the *TWP*. To inspect the spread of $\mathcal{I}_{\mathrm{s}}$ as a function of $R_{1240}$ for mixed-phase cases with different *IF*, either the $r_{\mathrm{eff}}$ of the liquid and ice particles, or the *TWP* were kept constant. The approach using a constant value of $r_{\mathrm{eff}}$ was evaluated for the two-layer (Fig. 8a) and the vertically homogeneous mixing scenarios (Fig. 8b), considering a fixed $r_{\mathrm{eff}}$ of $9\,\mu$m for
10  the liquid droplets and $50\,\mu$m for the ice crystals. The *TWP* was varied between $25\,\mathrm{g\,m^{-2}}$ and $250\,\mathrm{g\,m^{-2}}$. The fixed *TWP* approach was evaluated for the homogeneous mixing scenario (Fig. 8c). Here, the *TWP* was fixed to $120\,\mathrm{g\,m^{-2}}$. In this case,

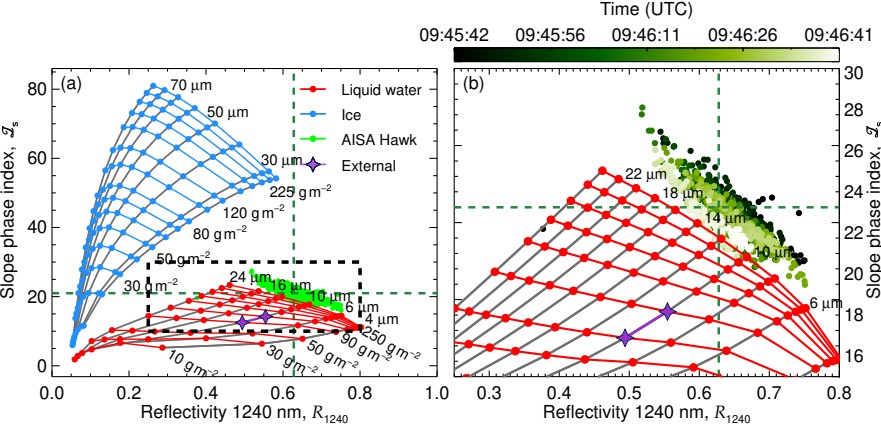

**Figure 7. (a)** $\mathcal{I}_\mathrm{s}$ measured on 2 June 2017 presented as a function of $R_{1240}$ (green dots). The dashed lines indicate the 25th percentile of $R_{1240}$ and the 75th percentile of $\mathcal{I}_\mathrm{s}$. The two grids represent radiative transfer simulations for a range of pure liquid (red) and pure ice (blue) clouds. The liquid water clouds cover droplets with $r_\mathrm{eff}$ between 4 and 24 $\mu$m and *LWP* between 1 and 250 g m$^{-2}$. The ice clouds are simulated for columnar ice crystals with $r_\mathrm{eff}$ between 28 and 90 $\mu$m and *IWP* between 1 and 250 g m$^{-2}$. A SZA of 57.9° was considered. The purple stars shows the independent *LWP* range retrieved by the 89 GHz passive channel of MiRAC and the SID-3 in situ observation of particle size. **(b)** Zoom into the area highlighted by a dashed rectangle in **(a)**. Color-coded is the acquisition time of measurements illustrating changes along the flight path.

the $r_\mathrm{eff}$ ranges between 4 $\mu$m and 24 $\mu$m for liquid droplets and between 28 $\mu$m and 90 $\mu$m for ice crystals. The three scenarios show grids of $\mathcal{I}_\mathrm{s}$ where the increasing *IF* yields different patterns. The comparison with the measurements shows that only the homogeneously mixed scenarios (Figs. 8b and 8c) may reproduce the measured values of the slope phase index. In the two-layers scenario (Fig. 8a), the liquid water signature dominates $\mathcal{I}_\mathrm{s}$, masking the presence of the cloud ice. These mixed-phase

5     clouds need to be formed of at least *IF* = 70 % to cause phase indices that effectively differ from those of pure liquid clouds. Additionally, the *TWP* required to match the observations exceeds the observed values. This indicates that a significant amount of ice near the cloud top is needed to explain the observed high values of $\mathcal{I}_\mathrm{s}$.

    The homogeneous phase mixing scenario presented on Fig. 8b could explain part of the observed values of the reflectivity and slope phase index. According to this scenario, the cloud holes (reflectivity below the 25th percentile of $R_{1240}$) would show

10    higher ice fractions (between 20 % and 40 %) and higher $\mathcal{I}_\mathrm{s}$ than the cloud dome centers (reflectivity above the 25th percentile of $R_{1240}$ and phase index below the 75th percentile of $\mathcal{I}_\mathrm{s}$), where *IF* is between 0 and 20 %. Figure 8c shows the alternative scenario where the *TWP* is fixed to 120 g m$^{-2}$. The simulated clouds cover most of the observed combinations of slope phase indices and reflectivities. In this scenario, the observed cloud would agree with mixed-phase clouds of fixed *IF* of about 40 %. In contrast to the scenario with fixed $r_\mathrm{eff}$, this pattern indicates that the ice fraction in the cloud centers is similar to that in

15    the cloud domes centers consist of small droplets with effective radii between 4 $\mu$m and 6 $\mu$m and small ice crystals with effective radii between 28 $\mu$m and 36 $\mu$m. Larger droplets, with $r_\mathrm{eff}$ between 6 $\mu$m and 8 $\mu$m, and ice crystals,

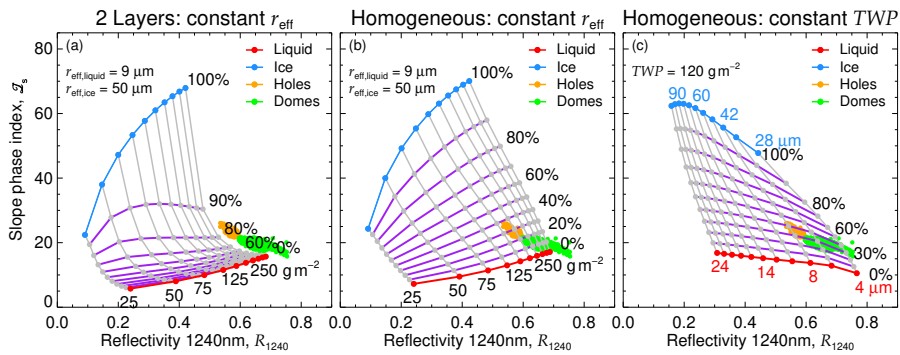

**Figure 8.** Comparison of $\mathcal{I}_s$ measured on 2 June 2017 as a function of $R_{1240}$ with three mixing scenarios of mixed-phase clouds. Observations in cloud holes are indicated by orange dots. Green dots represent measurements in cloud domes. Scenario (**a**) simulates a two layer cloud, while in scenarios (**b**) and (**c**) a homogeneously mixed cloud is assumed. Scenario (**b**) considers mixed-phase clouds of fixed particle sizes ($r_{\mathrm{eff,liquid}}$ of $9\,\mu m$ and $r_{\mathrm{eff,ice}}$ of $50\,\mu m$) and variable *TWP* between 25 and $250\,\mathrm{g\,m^{-2}}$. The grey solid lines connect clouds of equal *TWP* and the solid purple lines, clouds of equal *IF* (indicated by the percentages). In scenario (**c**) *TWP* is fixed to $120\ \mathrm{g\,m^{-2}}$ and the particle sizes are varied. Here, purple lines connect clouds of equal ice fraction and the gray lines connect clouds considering equal particle sizes.

with $r_{\mathrm{eff}}$ between $36\,\mu m$ and $42\,\mu m$ are found in the cloud holes. This pattern can be explained by a quick evaporation of small droplets in the cloud holes leading to a larger $r_{\mathrm{eff}}$. Both idealized homogeneous mixing scenarios reproduce the observations. However, based on the AISA Hawk measurements of $\mathcal{I}_s$ alone, it cannot be judged which scenario is more likely. In reality, neither the particle sizes nor the *TWP* are horizontally fixed in a cloud field. A combination of both scenarios might be closest to reality. However, due to the large number of possible realizations (combinations of *IWP*, *LWP*, $r_{\mathrm{eff,ice}}$, $r_{\mathrm{eff,liquid}}$), it is impossible to fully resemble the observations.

## 4 Comparison of measurements and LES

Comparing simulated cloud top reflectivies and phase index based on ICON-LEM cloud fields with the measurements of AISA Hawk will help to evaluate the conclusions about the vertical structure of the cloud thermodynamic phase drawn in the previous section.

For the two cloud cases of 25 May and 2 June, two regions of $21\,\mathrm{km} \times 11\,\mathrm{km}$ enclosing the corresponding aircraft measurements were simulated by ICON-LEM (Fig. 2). The resulting cloud profiles are shown in Figs. 9a - 9c, and 9e - 9g. The profiles of ice fraction *IF*(z) shown in Figs. 9b and 9f are calculated, in correspondence to Eq. 7, by:

$$IF(z) = \frac{IWC(z)}{LWC(z) + IWC(z)} \cdot 100\,\%. \tag{8}$$

On 25 May, the clouds simulated by ICON-LEM are located at higher altitudes than observed. However, the simulated profiles of *LWC*, *IWC*, and *IF* confirm the vertical cloud structure indicated by the active remote sensing measurements (Fig. 3a),

with both liquid and ice phases being present. The *IWC* reaches a maximum value of $0.08\,\mathrm{g\,m^{-3}}$ 430 m below the $0.12\,\mathrm{g\,m^{-3}}$ maximum *LWC* at 900 m.

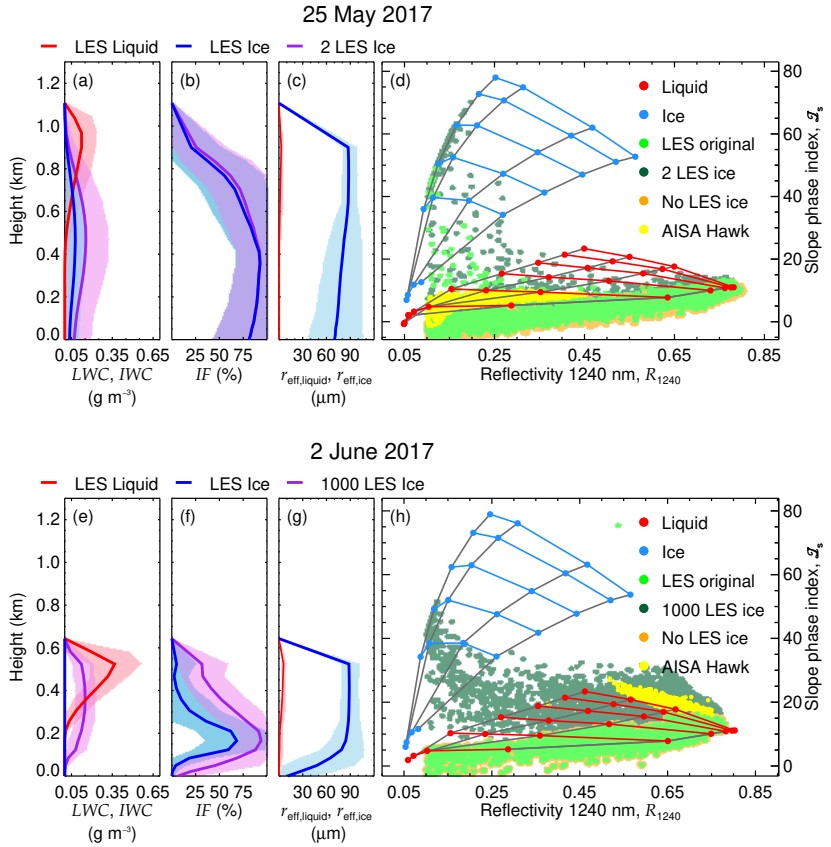

**Figure 9.** Mean profiles of liquid and ice water content, ice fraction and effective radius (**a**, **b**, and **c** for 25 May 2017 and **e**, **f**, and **g** for 2 June 2017, respectively). The shaded areas indicate the standard deviation of the considered distribution. The simulated $R_{1240}$ and $\mathcal{I}_\mathrm{s}$ corresponding to the original LES profiles, as well as simulations neglecting the *IWC* ('No LES ice') and modifying it ('2 LES ice' for 25 May and '1000 LES ice' for 2 June), are compared with $R_{1240}$ and $\mathcal{I}_\mathrm{s}$ of pure phase clouds and the AISA Hawk measurements in (**d**) (25 May) and (**h**) (2 June).

The cloud top reflectivities simulated by libRadtran on the basis of the clouds simulated by ICON-LEM have been used as synthetic measurements to calculate $\mathcal{I}_\mathrm{s}$. These synthetic $\mathcal{I}_\mathrm{s}$ are compared to the observations of AISA Hawk (Figs. 5 and 7).
To further test the sensitivity of $R_{1240}$ and $\mathcal{I}_\mathrm{s}$ towards the vertical distribution of the cloud thermodynamic phase, additional synthetic cloud top reflectivities (firstly, neglecting the simulated *IWC*, hence considering pure liquid water clouds, and secondly, doubling the simulated *IWC*), were also investigated. The comparisons with the AISA Hawk measurements is shown in Fig. 9d. The relation between $R_{1240}$ and $\mathcal{I}_\mathrm{s}$ derived from the LES original *LWC* and *IWC* profiles shows that the liquid

water dominated the cloud top layer, making its $R_{1240}$ and $\mathcal{I}_s$ indiscernible from those of pure liquid water clouds. This is almost identical to the AISA Hawk measurements (Fig. 9d). Only a few data points with higher $\mathcal{I}_s$ range above the grid of pure liquid water clouds. These data mostly have low $R_{1240}$ and can be linked to cloud edges with lower *LWP* located outside the measurement area of AISA Hawk, where ice fractions are simulated to be higher than observed. Doubling the simulated *IWC*
on 25 May (resulting in a maximum $0.16\,\mathrm{g\,m^{-2}}$ at 470 m) yielded a similar result: as for the originally simulated profiles, the $R_{1240}$ and $\mathcal{I}_s$ relation is for most LES pixels dominated by the higher liquid water concentration at cloud top and cannot be differentiated from pure liquid water clouds. However, the enhanced *IWC* increases $\mathcal{I}_s$ beyond values corresponding to pure liquid water clouds for a larger amount of cloud edge pixels than with the *IWC* originally simulated by ICON-LEM.

On 2 June, ICON-LEM produces a maximum *IWC* of $1.5{\times}10^{-4}\,\mathrm{g\,m^{-3}}$ located 170 m below the maximum $0.37\,\mathrm{g\,m^{-3}}$
*LWC* at 530 m. As for 25 May, the vertical profiles of *IWC* and *LWC* agree with the active remote sensing measurements (Fig. 3b), indicating the presence of both liquid and ice. However, as demonstrated by Fig. 9h, the original *IWC* simulated by ICON-LEM is too low to effectively impact $R_{1240}$ and $\mathcal{I}_s$, which follow the pattern of pure liquid water clouds and did not reproduce the AISA Hawk observations. This difference suggests that the ICON-LEM underestimates the concentration of ice for the cloud on 2 June 2017. In a test case, the *IWC* was increased by a factor of 1000 (maximum value of $1.5{\times}10^{-4}\,\mathrm{g\,m^{-3}}$
at 360 m) in the same order of magnitude than the maximum *LWC*. For this hypothetical cloud field, the radiative transfer simulations reproduced the observed values of $\mathcal{I}_s$, which deviate from the pure liquid case. However, the results of the ICON-LEM simulations show many data points with $R_{1240}$ way below the observations ($R_{1240} < 0.45$). This indicates that the cloud field produced by the LES, covering a larger area than the observations, presents significant cloud gaps (low *TWP*), which were located outside the AISA Hawk measurement region. For the manipulated cloud, these cloud parts show a significant increase
of $\mathcal{I}_s$ with decreasing $R_{1240}$, which can be attributed to cloud edges similar to the cold air outbreak case of 25 May.

## 5   Impact of spatial resolution

The horizontal resolutions of the ICON-LEM (100 m) and the airborne observations (10 m) differ by about one order of magnitude. Additionally, satellite-borne imaging spectrometers commonly used to derive global distributions of cloud properties typically do not reach a spatial resolution as high as the AISA Hawk measurements. For instance, the Advanced Very High
Resolution Radiometer (AVHRR), the MODerate resolution Imaging Spectroradiometer (MODIS), and the Hyperion imaging spectrometer have resolutions of 1000 m, 500 m, and 30 m pixel sizes, respectively (Kaur and Ganju, 2008; Li et al., 2003; Thompson et al., 2018). This raises the question of how much of the observed variability of $\mathcal{I}_s$ is lost by horizontal averaging. To asses this question, the AISA Hawk observations of the two cloud cases were averaged for larger pixel sizes. Figures 10 and 11 show a 900 m × 900 m subsection of the original fields of $R_{1240}$ and $\mathcal{I}_s$ projected for pixel sizes of 30 m (Hyperion), 90 m
(~ICON-LEM), 450 m (~MODIS), and 900 m (~AVHRR). The relationship between $\mathcal{I}_s$ and $R_{1240}$ of the complete fields is illustrated in Figs. 10c, 10f, 10i, 10l, and 10o for 25 May 2017 and in Figs. 11c, 11f, 11i, 11l, and 11o for 2 June 2017. The statistics of $R_{1240}$ and $\mathcal{I}_s$ corresponding to the considered pixel sizes for both days are presented in Tab. 2.

**Table 2.** $R_{1240}$ and $\mathcal{I}_\mathrm{s}$ dependence upon the sensor resolution.

| | | 25 May 2017 | | | | 2 June 2017 | | | |
|---|---|---|---|---|---|---|---|---|---|
| | | Min. | Max. | 25th percentile | 75th percentile | Min. | Max. | 25th percentile | 25th percentile |
| $R_{1240}$ | Original | 0.10 | 0.50 | 0.16 | 0.28 | 0.18 | 0.83 | 0.63 | 0.68 |
| | 30 m | 0.10 | 0.48 | 0.16 | 0.28 | 0.45 | 0.76 | 0.63 | 0.68 |
| | 90 m | 0.10 | 0.42 | 0.16 | 0.28 | 0.51 | 0.72 | 0.63 | 0.67 |
| | 450 m | 0.13 | 0.33 | 0.17 | 0.22 | 0.63 | 0.67 | 0.64 | 0.66 |
| | 900 m | 0.14 | 0.23 | 0.15 | 0.20 | 0.64 | 0.66 | 0.64 | 0.65 |
| $\mathcal{I}_\mathrm{s}$ | Original | -2.12 | 11.7 | 6.54 | 8.29 | 15.0 | 36.3 | 19.1 | 21.0 |
| | 30 m | 0.07 | 9.90 | 6.60 | 8.23 | 16.5 | 29.8 | 19.3 | 20.9 |
| | 90 m | 3.45 | 9.43 | 6.62 | 8.08 | 17.7 | 25.0 | 19.5 | 20.7 |
| | 450 m | 5.54 | 8.15 | 6.94 | 7.67 | 19.0 | 20.9 | 19.3 | 20.4 |
| | 900 m | 6.60 | 7.71 | 6.77 | 7.13 | 19.1 | 19.9 | 19.5 | 19.7 |

The smoothing of the cloud scene with increasing pixel size erases the fine spatial structure of the cloud top, which remains only visible for 25 m pixel size. For the cloud case of 25 May 2017, the horizontal averaging mainly impacts the observed cloud geometry. The decreasing contrast between the cloudy and cloud-free pixel changes the cloud mask and eventually causes the loss of the cloud broken nature observed by AISA Hawk. The original range of variability of $R_{1240}$ between 0.10 and 0.50 decreases to the range between 0.14 and 0.23 at 900 m. The original range of $\mathcal{I}_\mathrm{s}$ between -2.12 and 11.7 is reduced to the range from 6.60 to 7.71, but always indicates a cloud that is dominated by the liquid layer at cloud top. For the cloud on 2 June 2017 (Fig. 11), the averaging cannot affect the 100 % cloud cover. However, the variability of $R_{1240}$ becomes significantly reduced for larger pixel sizes (from the original variability between 0.18 and 0.83 to a variability at 900 m between 0.64 and 0.66) as no large-scale cloud structures are present. Similarly, the variability of $\mathcal{I}_\mathrm{s}$ diminishes for observations with coarser spatial resolution from the original range between 15.0 and 36.3 to 19.1 and 19.9 for pixels of 900 m). A coarser resolution removes the contrast between cloud holes, which are typically characterized by the presence of ice crystals (high $\mathcal{I}_\mathrm{s}$) and the cloud domes, where liquid droplets dominate (lower $\mathcal{I}_\mathrm{s}$). For satellite observations with pixel sizes larger than 90 m, this prevents from characterizing and interpreting the change of cloud phase in the small scale cloud structure and, therefore, conceals the information about the vertical distribution of the thermodynamic phase contained in the cloud top variability. High resolved imaging spectrometer measurements such as the Hyperion and the ICON-LEM, with pixels below 100 m are still able to resolve part of the natural horizontal variability.

## 6 Conclusions

Based on airborne active and passive remote sensing conducted by a passive imaging spectrometer and vertically resolving instruments, such as lidar and radar, the horizontal and vertical structure of the thermodynamic phase in Arctic mixed-phase

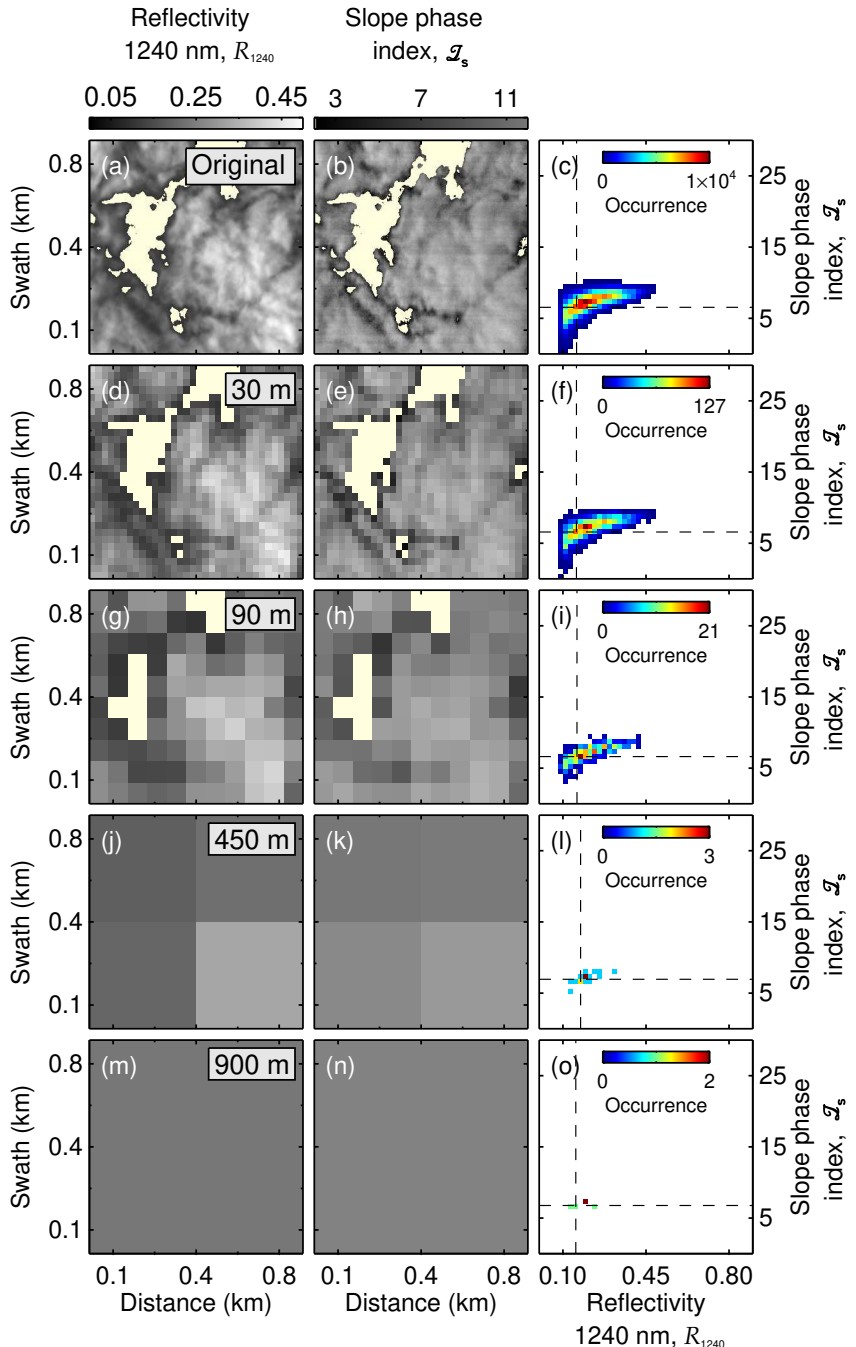

**Figure 10.** Slope phase index - 1240 nm reflectivity relationship for 5 different pixel sizes (original AISA Hawk resolution, 30 m, 90 m, 450 m, and 900 m). (**a**), (**d**), and (**g**) show a 1 km × 1 km subsection of $R_{1240}$ measured on 25 May 2017 as seen by the five different resolutions; (**b**), (**e**), and (**h**), the corresponding 1 km × 1 km $\mathcal{I}_s$; and (**c**), (**f**) and (**i**) present the scatter between both magnitudes for the complete 1 km × 4 km field. The dashed lines indicate the 25th percentile of $R_{1240}$ and $\mathcal{I}_s$ for each resolution.

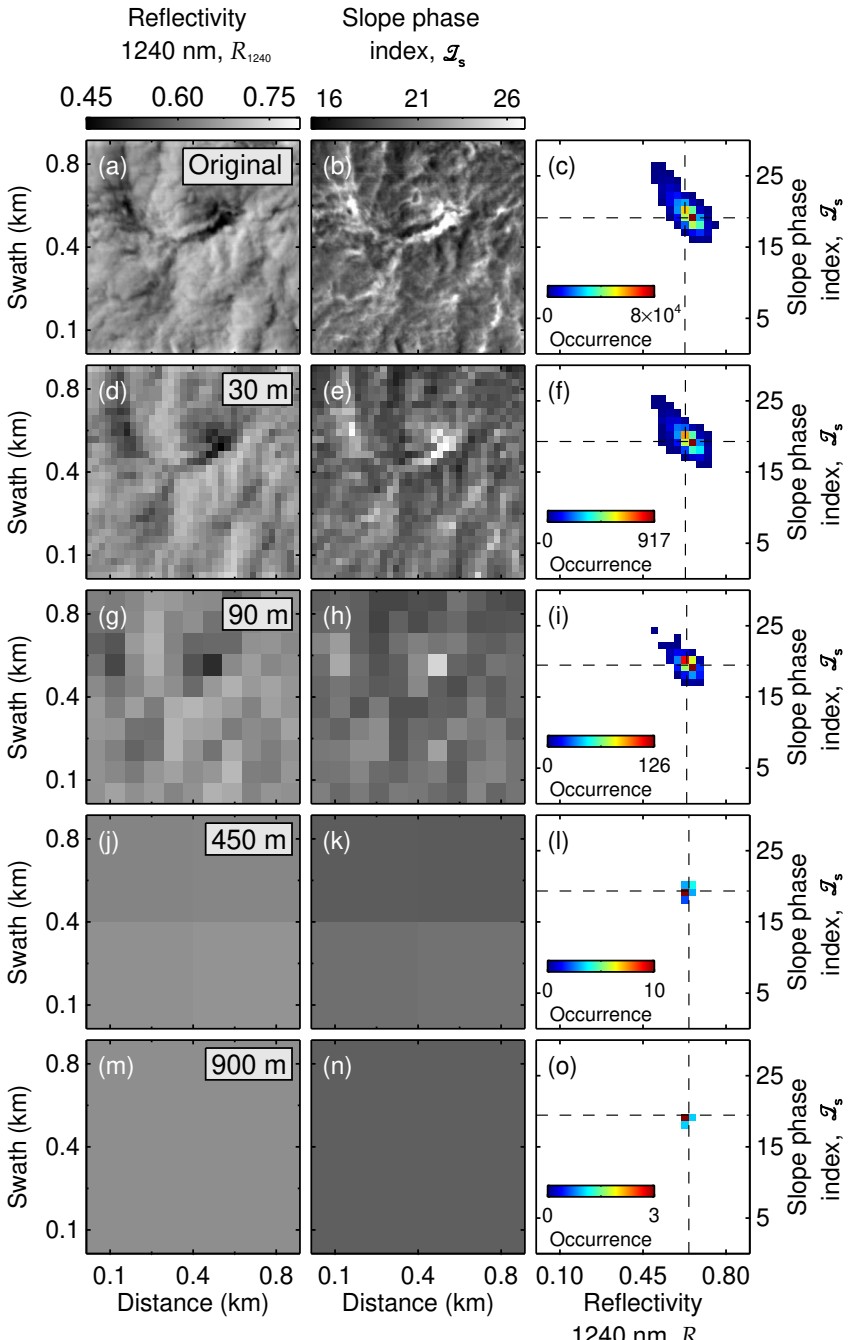

**Figure 11.** Slope phase index - 1240 nm reflectivity relationship for 5 different pixel sizes (original AISA Hawk resolution, 30 m, 90 m, 450 m, and 900 m). (**a**), (**d**), and (**g**) show a 1 km × 1 km subsection of $R_{1240}$ measured on 2 June 2017 as seen by the five different resolutions; (**b**), (**e**) and (**h**), the corresponding 1 km × 1 km $\mathcal{I}_s$; and (**c**), (**f**) and (**i**) present the scatter between both magnitudes for the complete 1 km × 4 km field. The dashed lines indicate the 25th percentile of $R_{1240}$ and $\mathcal{I}_s$ for each resolution.

cloud cases was characterized for two example clouds observed during a cold air outbreak and a warm air intrusion event. While the spectral imaging was used to identify the structure of the horizontal distribution of the cloud ice at scales down to 10 m, the combined radar and lidar observations revealed the general vertical thermodynamic phase distribution of the clouds.

The two cloud cases were observed over open ocean close to Spitzbergen during the ACLOUD campaign. The cloud scene sampled on 25 May 2017 evolved within a cold air outbreak, whereas a cloud that had formed in a warm air advection event was sampled on 2 June 2017. For both cloud cases, the combined radar and lidar observations indicated the mixed-phase character of the clouds, with liquid water droplets in the cloud top layer and ice crystals below. While the lidar penetrated the strongly reflecting liquid cloud layer on 25 May, partly until the surface, the strong extinction of the lidar signal close to the cloud top observed on 2 June indicates higher liquid water amounts. The vertical structure of the radar backscatter also differs between both days, with reflectivities reaching the ground on 25 May typical for light snow precipitation. These different cloud vertical structures influenced the ability to detect the ice by the imaging spectrometer observations of AISA Hawk using the slope phase index $\mathcal{I}_s$. On 25 May, $\mathcal{I}_s$ is dominated by the liquid water contained at the cloud top layer, which leads to a misclassification as a pure liquid water cloud. The small-scale variability of $\mathcal{I}_s$ observed on 25 May relates mostly to the variability of the liquid cloud layers. On 2 June, AISA Hawk measured higher $\mathcal{I}_s$, which hints at the presence of ice crystals in higher cloud layers. Additionally, the *LWP*, retrieved by assuming pure liquid clouds, shows unrealistically high values compared to the observations by MiRAC, which supports this conclusion. The high values of $\mathcal{I}_s$ and the large retrieval bias of *LWP* are observed close to areas of low cloud reflectivity (cloud holes). The comparison of both cloud cases highlights the limitations of passive remote sensing alone to identify layered mixed-phase structures if the ice is not sufficiently close to the cloud top. In particular in these cases, the combination of active and passive remote sensing is crucial to fully characterize the horizontal and vertical distribution of ice and liquid water particles in mixed-phase clouds.

The highly resolved horizontal distribution of $\mathcal{I}_s$ observed on 2 June was analyzed using radiative transfer simulations assuming different mixing scenarios of ice and liquid water content. Two homogeneous mixing scenarios, either keeping the *TWP* or the particle sizes fixed when changing the ice fraction, did reproduce the observed pattern of variability. However, based on the AISA Hawk measurements of $\mathcal{I}_s$ alone, it cannot be judged which scenario is closer to reality. To consider modeled phase-mixing scenarios of *IWP*, *LWP*, $r_{\text{eff,ice}}$, $r_{\text{eff,liquid}}$ and the vertical cloud structure, the ICON-LEM was applied. The microphysical profiles simulated by ICON-LEM roughly represent major features of the vertical profiles obtained by MiRAC and AMALi for both cloud cases. To compare with the AISA Hawk measurements, radiative transfer simulations of the cloud top were performed on the basis of the ICON-LEM thermodynamic phase profiles. For both cases, the variability of $\mathcal{I}_s$ calculated from the simulations is represented by pure liquid water clouds. Enhancing the *IWC* simulated by ICON-LEM indicates that, whereas on 25 May this behavior is due to the liquid-water-dominated cloud top layer, on 2 June, the simulated concentration of ice crystals is underestimated. In a test case where the $IWC$ was enhanced 1000 times, the simulated cloud central regions showed a comparable structure as observed by AISA Hawk. Additionally, the area simulated by ICON-LEM produced significant cloud gaps not present in the smaller cloud section observed by AISA Hawk. Similarly to 25 May, the cloud gaps present high values of $\mathcal{I}_s$. The comparison of the simulated $\mathcal{I}_s$-$R_{1240}$ patterns with measured ones can be used used to assess the performance of ICON-LEM, which reproduces the vertical structure of the two observed cloud cases, but produces

too little ice on 2 June. Nevertheless, to fully exploit the measurements-model synergy, synthetic radar and lidar measurements should be simulated based on ICON-LEM, taking as well into consideration the ice habit observed by in-situ measurements.

The grid size of ICON-LEM (100 m) is sufficient to resolve the small-scale structure of mixed-phase clouds and to produce different patterns of $\mathcal{I}_\mathrm{s}$ giving indication on the vertical distribution of the cloud thermodynamic phase. A sensitivity study reducing the horizontal resolution of the passive remote sensing observations illustrated that pixel sizes below 100 m, such as provided by the Hyperion imager spectrometer or airborne spectral imagers, are required to resolve the horizontal distribution of ice and liquid water in Arctic mixed-phase clouds. However, common satellite sensors such as MODIS or AVHRR are not able to capture the small-scale distribution of $\mathcal{I}_\mathrm{s}$.

*Data availability.* The AISA Hawk (Ruiz-Donoso et al., 2019), SMART (Jäkel et al., 2019), MiRAC (Kliesch and Mech, 2019) and AMALi (Neuber et al., 2019) data, acquired during the ACLOUD campaign, are publicly available on PANGAEA. All other data used and produced in this study are available upon request from the corresponding authors.

## Appendix A: Analysis of 3D effects in the AISA Hawk measurements

The spatially highly-resolved radiance fields measured by AISA Hawk are affected by 3D radiative effects caused by the three-dimensional nature of the cloud top. Specifically, a) horizontal photon transport occurs between neighboring pixels, smoothing the measurements, and b) cloud top structures cast shadows on the image.

In order to correct the smoothing due to horizontal photon transport, the horizontal sensitivity of each case study was estimated comparing 3D and 1D simulations of th cloud top reflected radiance. The 3D simulations of an idealized cloud field were performed with the Monte Carlo Atmospheric Radiative Transfer Simulator (MCARaTS, Wang et al., 2012) and the 1D simulations were performed with libRadtran. The cloud field considers a liquid water stratiform deck with a $LWP$ similar to the observations (i.e. $30\,\mathrm{g\,m^{-2}}$ and $100\,\mathrm{g\,m^{-2}}$, respectively), a typical $r_\mathrm{eff}$ of $10\,\mu$m and solar zenith angle (SZA) of $60°$ and $57°$, respectively. A pure ice region of 15 m width, with $r_\mathrm{eff}$ of $60\,\mu$m and a $IWP$ similar to the $LWP$, was embedded in the liquid deck. The change of cloud phase in general leads to a reduction of the cloud top radiance in the ice phase area. The 3D and 1D simulations of the $100\,\mathrm{g\,m^{-2}}$ case are presented in Fig. A1a. Whereas the 1D simulated radiance stays constant in the liquid water region and decreases sharply within the ice stripe, the horizontal photon transport smooths the transition from the liquid to the ice region in the 3D radiance. The cross-correlation between both simulations, shown in Fig. A1b, provides an estimation of the horizontal displacement of the photons in the 3D simulation, which is effective within distances of about 100 m. The combination of cross-correlation functions calculated for different solar azimuth angles, SAA, and different sensor viewing angles (therefore accounting for different sun-sensor geometries) yields the three dimensional normalized convolution kernel $CK$ presented in Fig. A1c. The simulations with $LWP$ of $30\,\mathrm{g\,m^{-2}}$ (not shown here) yield a similar result. The derived $CK$ accounts only for the mean photon transport of each field and does not consider local inhomogeneities. Similar to Zinner

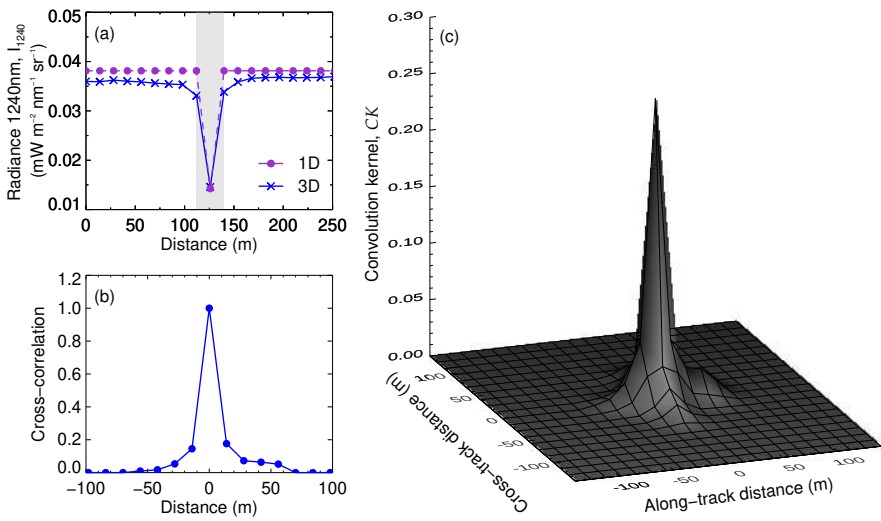

**Figure A1.** (**a**) Comparison of the nadir reflected radiance at 1240 nm by a stratiform cloud deck simulated with 1D and 3D radiative transfer simulations. The cloud contains a 15 m region of pure ice (shaded) embedded between two pure liquid water regions (non-shaded). (**b**) Cross-correlation between the 1D and the 3D cloud top radiance illustrating the extent of the horizontal photon transport. (**c**) Normalized convolution kernel based on the cross-correlation of the 1D and 3D simulations and different sun-sensor geometries.

et al. (2006), in order to avoid overcompensating the horizontal photon transport, the iterative Richardson-Lucy deconvolution algorithm (Richardson, 1972; Lucy, 1974) was applied. After each iteration, the calculated radiance takes the form,

$$I_{n+1} = \left[ \left( \frac{I}{I_n \otimes CK} \right) \otimes CK \right], \tag{A1}$$

where $I$ is the radiance observed by AISA Hawk, $I_n$ is the radiance obtained after the $n^{th}$ iteration, and $\otimes$ is the convolution operator. Based on the convergence of $|I_{n+1} - I_n|/I_n$, a number of 4 iterations was found to sufficiently increase the sharpness of the measured radiance fields.

However, the second 3D radiative effect, caused by the shadows casted by the cloud top geometry, cannot be easily corrected. Highly spatially resolved measurements of the cloud top geometry would be necessary for correcting self-shading artifacts. Therefore, 3D radiative transfer simulations are used to estimate this 3D radiative effect and analyze whether the observed correlation between $R_{1240}$, $\mathcal{I}_s$, and $LWP$ are caused by shadows or by inhomogeneous distributions of the cloud thermodynamic phase. Figure A2 presents 3D simulations of two idealized stratiform cloud decks with a constant $TWP$ of $100\,\mathrm{g\,m^{-2}}$.

Figure A2a represents a liquid water cloud with an inhomogenous cloud top height (50 m lower cloud top in the center of the cloud field). For a SZA of 57°, similar to the measurement on 2 June, the dip on the cloud top casts a shadow that gets imprinted on $R_{1240}$ (A2c), $\mathcal{I}_s$ (A2e) and the retrieved $LWP$(A2g). Whereas in the shaded region $R_{1240}$ decreases on average

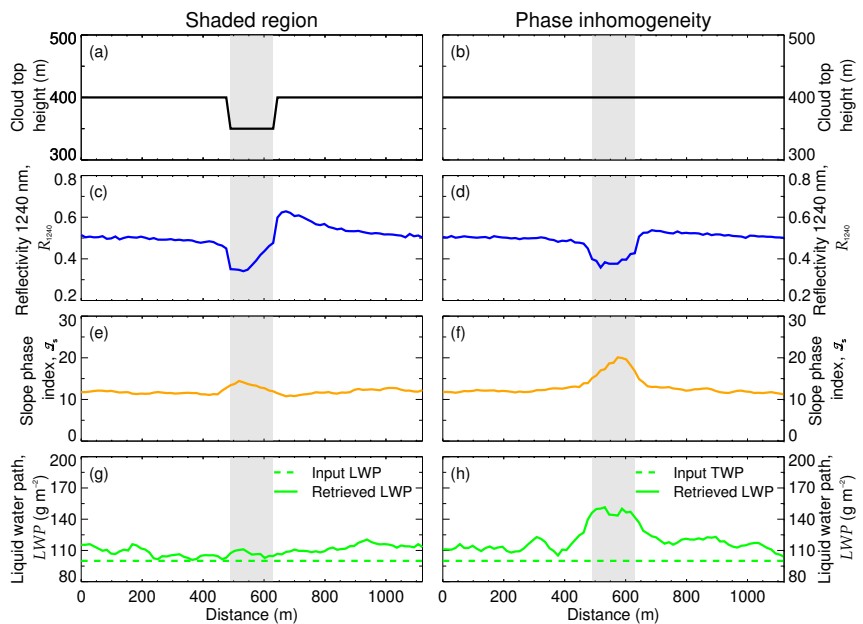

**Figure A2.** Cloud top properities of a shaded region (**a**, **c**, **e**, **g**) compared to a region with a different thermodynamic phase composition (**b**, **d**, **f**, **h**). The shaded areas indicates the artifact affected areas

by 35 % with respect to the non-shaded region, $\mathcal{I}_s$ increases on average by 20 %. These opposite effects result in an almost constant $LWP$, which does not show a signature of the cloud dip.

Figure A2b shows a pure liquid water cloud with a constant cloud top height and an embedded mixed-phase region of 150 m horizontal extent. The $TWP$ is kept always constant at 100 g m$^{-2}$ (i.e. the pure phase region considers a $LWP$ of 100 g m$^{-2}$; 5 the mixed-phase region considers a $LWP$ of 60 g m$^{-2}$ and a $IWP$ of 40 g m$^{-2}$). The liquid water droplets have an $r_{\text{eff}}$ of 10 $\mu$m and the ice crystals have an $r_{\text{eff}}$ of 60 $\mu$m. The inhomogeneous phase distribution obviously biases the retrieved cloud top properties and the calculated phase index. In this case, $R_{1240}$ (A2d) decreases by 34 % in the mixed-phase region compared to the pure-phase region, $\mathcal{I}_s$ increases by 58 %. However, constrasting the shaded case, the presence of ice crystals lead to a significant increase of $LWP$ by 36 %.

10 Therefore, the combination of $R_{1240}$, $\mathcal{I}_s$ and $LWP$ is crucial to interpret the observations of AISA Hawk. Only a simultaneous increase in $\mathcal{I}_s$ and $LWP$ when $R_{1240}$ decreases is indicative of mixed-phase regions. Although we cannot completely discard shading artifacts on the 2 June case study, the observed increment of $\mathcal{I}_s$ and $LWP$ in regions of low $R_{1240}$ agrees with the simulations in Figs. A2d, A2f and A2h and support the hypothesis of mixed-phase on this day.

## Appendix B: *LWP* retrieval based on passive microwaver radiometer measurements

Measurements by the 89 GHz passive channel of the Microwave Radiometer for Arctic Clouds (MiRAC, Mech et al., 2019) were used to estimate the liquid water path (*LWP*) for the two case studies. Brightness temperatures (TB) were measured under a tilted angle of 25° with respect to nadir backwards with 1 s integration time. At this frequency, TB depends on the surface emission, dependent in turn on the sea surface temperature (SST) and wind speed, and on atmospheric contributions by atmospheric gases and cloud liquid. Cloud ice does not contribute to the signal and only strong snowfall could lead to TB reduction by scattering, i.e. $500\,\mathrm{g\,m^{-2}}$ snowfall correspond to about 1-2 K reduction. On short time scales – such as the two minute long flight tracks – variations are mainly caused by cloud variability. Therefore, a simplified algorithm exploiting the relative change of TB compared to a base state was developed.

For each of the two cases, the closest dropsonde was used to calculate TB as a function of *LWP*, assuming a cloud between 500 and 100 m above sea level. Within these microwave radiative transfer simulations, the wind speed was taken from the lowest available dropsonde level ($5\,\mathrm{m\,s^{-1}}$ on 25 May and $7.7\,\mathrm{m\,s^{-1}}$ on 2 June) and the SST (275 K) from climatological data. Liquid water emission leads to an increase in TB above the radiatively cold ocean. When subtracting the clear sky TB ($\mathrm{TB_0}$), the resulting $\Delta$TB can be well approximated by a third order regression with an uncertainty of ca. $1\,\mathrm{g\,m^{-2}}$ in *LWP*. Due to the different wind speed and moisture conditions of the two cases, uncertainties of about $5\,\mathrm{g\,m^{-2}}$ ($12\,\mathrm{g\,m^{-2}}$) at $100\,\mathrm{g\,m^{-2}}$ ($200\,\mathrm{g\,m^{-2}}$) *LWP* occur.

The clear sky $\mathrm{TB_0}$ needs to be derived before applying the simple regression algorithm to calculate $\Delta$TB. For this purpose, we searched for the minimum TB in both cases and checked whether the lidar signal was low. This is to some degree subjectively and difficult due to the high cloud presence (see Figs. 4 and 6). In fact, for 2 June a profile approximately 5 min later was chosen. With our best estimates of $\mathrm{TB_0}$ (180 K on 25 May and 186 K on 2 June) for each one second measurement, *LWP* could be derived, yielding a range between 20 and $40\,\mathrm{g\,m^{-2}}$ for 25 May and 90 to $120\,\mathrm{g\,m^{-2}}$ for 2 June.

While the approach to derive *LWP* from a single frequency is rather simple, it also presents advantages (for example, absolute calibration errors are avoided due to the use of difference values). Changes in SST, wind speed and moisture content of the two one-minute time periods are thought to play a minor role and estimated to be below 10%. The highest uncertainty is thought to stem from the determination of the clear sky $\mathrm{TB_0}$. However, the maximum uncertainty is estimated to be about $30\,\mathrm{g\,m^{-2}}$ and thus, the 2 June case clearly (i) has a higher *LWP* than the 25 May case and (ii) has a lower *LWP* than the one estimated by AISA Hawk (Tab. 1). In the future, additional measurements from higher MiRAC frequency channels and lidar information will be exploited to retrieve a higher accuracy LWP product.

*Author contributions.* MW, AE and SW designed the experimental basis of this study. ERD, MS and EJ acquired the measurements of AISA Hawk and SMART. ERD selected the case studies, processed and analyzed the measurements of AISA Hawk, performed the 1D radiative transfer simulations, and drafted the manuscript. EJ processed the measurements of SMART and performed the 3D radiative transfer simulations. MM and RN acquired the measurements of MiRAC and AMALi. SC, MM, BSK, LLK and RN processed and analyzed

the measurements of MiRAC and AMALi. VS performed the ICON-LEM simulations. AE, MS and MW provided technical guidance. All authors contributed to the editing of the manuscript and to the discussion and interpretation of the results.

*Competing interests.* The authors declare that they have no conflict of interests.

*Acknowledgements.* We gratefully acknowledge the funding by the Deutsche Forschungsgemeinschaft (DFG, German Research Foundation) – Projektnummer 268020496 – TRR 172, within the Transregional Collaborative Research Center "ArctiC Amplification: Climate Relevant Atmospheric and SurfaCe Processes, and Feedback Mechanisms (AC)[3].

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
