# Peer review of "Small-scale structure of thermodynamic phase in Arctic mixed-phase clouds observed by airborne remote sensing during a cold air outbreak and a warm air advection event"

_Atmospheric Chemistry and Physics, 2019_

## Referee Comment (RC1) · David R. Thompson (Referee) · 9 Nov 2019

The authors provide a well-motivated study into the small-scale vertical and horizontal structure of cloud thermodynamic phase. The experiment is impressive in scope, including both passive (horizontal mapping) and active (vertical) airborne instruments, combined with additional large eddy simulations of vertical structure. The authors conclude that, given their data, "the cloud top small-scale horizontal variability reacts to changes in the vertical distribution of the cloud thermodynamic phase." This is an important topic for GCM parameterization. The manuscript is very clear and well-written,

though I have identified some potential weaknesses in both the methodology and scope that the authors might consider.

Overall: I am sympathetic to, and appreciative of, the authors' attention to fine-scale thermodynamic phase. However, such measurements may eventually require a more quantitative and comprehensive account for other potential confounding variance at scales of 100m or less. Taking this on would significantly improve the paper.

1. 2D photon transport effects in the cold outbreak case. As far as I can tell from the description, the authors' phase index retrieval method assumes RTMs of a homogeneous plane-parallel cloud surface; this is fine, except that they then apply that to interpret a heterogeneous cloud body where there is likely to be considerable horizontal photon transport due to scattering within the cloud itself. Horizontal transport implies the retrieval has a locale-specific geometric sensitivity "footprint" which I suspect is larger than the 10m spatial resolution of AISA. But the authors' maps seem to interpret every pixel at native resolution as if it were an effective discriminator of cloud phase at that location - hence the frequency distribution histograms of Fig. 5. Additionally, the exposure to incident sunlight and slant path through the cloud might vary since the cloud thickness and altitude are also varying on similar scales (i.e. "Domes" and "Holes" but also even finer-scale structure in the retrievals)

   While a full 3D simulation may be overkill, the authors would ideally find sufficient account for the cloud horizontal heterogeneity in their algorithm to avoid hallucinating compositional features which might, for example, be artifacts of self-shading effects or other horizontal heterogeneity. As a thought experiment, imagine where a reasonable estimate of horizontal sensitivity could be determined and the AISA data were convolved with a sensitivity kernel before applying the retrieval. If this caused the LWP retrieval to look more like the 100+ m rows of Figs. 10 and 11, how would it affect the authors' top-level claims and interpretations of

fine-scale structure?

2. Accuracy of the slope phase index for characterizing mixed phase clouds. While $\mathcal{I}_s$ has been demonstrated as an effective way to discriminate pure clouds, it is not always clear how to interpret $\mathcal{I}_s$ for mixed phase clouds since, as the authors note, the vertical partitioning can favor one or the other. In fact, the particles can themselves be mixed together at *very* small spatial scales - such intimate mixing could contribute to the strong overestimation of LWP for the warm air advection scenario (as noted at the bottom of page 11). Even the interpretation as intimately-mixed particles is suspect though since the "fishnet" manifold is so tight near the AISA data in Fig. 8. Does the ratio really provide enough measurement power to discriminate mixed phases, particularly given the uncharacterized uncertainties I mention above - how can we exclude self-shading and sky view fractions as an alternative account for the apparent difference between "Holes" and "Domes?"

3. Formal hypothesis testing, or uncertainty quantification or propagation. The general character of the manuscript is to bring together multiple measurement modes (LWP by AISA and MiRAC) and conclude that both are important to interpret diverse cloud structure. Fair enough, but do the authors have an uncertainty budget for either instrument? For example, what is the error in the Backscatter - the vertical profiles seem very qualitative in nature and it is unclear to what degree the temporal axis represents meaningful change in structure. This is important in comparing said structure to the LWP maps - which are themselves uncertain up to a level determined by the instrument noise and unknowns in the observation system. LES analysis is of a similarly qualitative nature, and an important first step which is meritorious as an exercise, but given the large mismatch between distributions of simulated and measured $\mathcal{I}_s$ it is not clear that the LES has successfully explained or even accounts for the observations. Can the authors formulate this as a formal hypothesis test of some kind?

[Figure]

Minor technical clarification: How do the horizontal (i.e.) spatial scales compare between Fig. 3 and the AISA data?

---

## Referee Comment (RC2) · Anonymous Referee #2 · 6 Jan 2020

Review of "Small-scale structure of thermodynamic phase in Arctic mixed-phase clouds observed by airborne remote sensing during a cold air outbreak and a warm air advection event" by Ruiz-Donoso et al.

Recommendation: Accept with minor revision

This paper uses a combination of active and passive remote sensing instrumentation to characterize the small scale structure of cloud thermodynamic phase using two case studies observed in the Arctic. The paper is well written, gives significant results and

the approach seems technically sound. Thus, as such, I think it is appropriate for publication in ACP. However, there are a few minor editorial comments and some additional points that the authors might want to consider before the publication is finalized.

I would like to see more explanation on why the two particular case studies were chosen and how representative these case studies are of conditions encountered in the Arctic in general. Although cases of single-layer mixed-phase clouds do occur in the Arctic as the authors state, and although they are nice to examine from a process-oriented perspective because it involves the complications of interactions between different cloud layers, past studies have suggested that multi-layer clouds and even multi-layer mixed-phase clouds may be more common than these single-layer clouds. Thus, some explanation of how the results from these special cases are applicable in general to remote sensing (especially cases when active remote sensing data are not available) would be appropriate.

Were there any Doppler radar data available? Some past studies have shown that the presence of cloud top generating cells frequently occur in the Arctic (as well as in other regions) and could be responsible for some of the horizontal inhomogeneity. If such data are available, perhaps more could be said about the scales of mixing of the phases and their horizontal distributions (and the processes). This would also give more information about the resolution required for analysis.

My other concern relates to the use of hexagonal columns to characterize the ice crystals. A lot of previous studies have suggested that the majority of ice crystals in Arctic clouds, including those in mixed-phase conditions, are very irregular and not well characterized by pristine shapes. Can a more realistic assumption about the ice crystal shapes be used? Or, alternatively, there should be more discussion made about the quantitative uncertainties induced by this simplistic assumption.

Page 4, line 30: It is not true in general that the radar reflectivity for ice is proportional to the sixth power of the ice particle size. For example, Hogan and collaborators

have developed much better quantitative models for converting ice crystal particle size distributions to radar reflectivity.

Page 5, line 8: How can the standard profile be used in combination with the dropsonde data? Wouldn't one or the other have to be used to give the vertical profile?

Page 6, line 3: What are unrealistic values of LWP? This should be more quantitative.

Page 6, line 6: What quantitative criteria were used to identify the presence of cold air outbreaks?

Page 7, line 6: Why couldn't it also be attributed to a reduced concentration of particles rather than just small particles?

---

## Author Comment (AC1) · 25 Mar 2020

**Answers to David R. Thompson (Referee 1)**

**General comments**

The authors provide a well-motivated study into the small-scale vertical and horizontal structure of cloud thermodynamic phase. The experiment is impressive in scope, including both passive (horizontal mapping) and active (vertical) airborne instruments, combined with additional large eddy simulations of vertical structure. The authors conclude that, given their data, "the cloud top small-scale horizontal variability reacts to changes in the vertical distribution of the cloud thermodynamic phase." This is an important topic for GCM parameterization. The manuscript is very clear and well-written, though I have identified some potential weaknesses in both the methodology and scope that the authors might consider. Overall: I am sympathetic to, and appreciative of, the authors' attention to fine-scale thermodynamic phase. However, such measurements may eventually require a more quantitative and comprehensive account for other potential confounding variance at scales of 100m or less. Taking this on would significantly improve the paper.

We thank the reviewer for his careful reading of the manuscript and his suggestions. With our reply and the revised version of our manuscript, we hope to address all the comments in a satisfying way. Our answers are structured and indicated as follows: reviewer comments (bold), answers, changes in the manuscript (italic).

1. **2D photon transport effects in the cold outbreak case. As far as I can tell from the description, the authors' phase index retrieval method assumes RTMs of a homogeneous plane-parallel cloud surface; this is fine, except that they then apply that to interpret a heterogeneous cloud body where there is likely to be considerable horizontal photon transport due to scattering within the cloud itself. Horizontal transport implies the retrieval has a locale-specific geometric sensitivity "footprint" which I suspect is larger than the 10m spatial resolution of AISA. But the authors' maps seem to interpret every pixel at native resolution as if it were an effective discriminator of cloud phase at that location - hence the frequency distribution histograms of Fig. 5. Additionally, the exposure to incident sunlight and slant path through the cloud might vary since the cloud thickness and altitude are also varying on similar scales (i.e. "Domes" and "Holes" but also even finer-scale structure in the retrievals). While a full 3D simulation may be overkill, the authors would ideally find sufficient account for the cloud horizontal heterogeneity in their algorithm to avoid hallucinating compositional features which might, for example, be artifacts of selfshading effects or other horizontal heterogeneity. As a thought experiment, imagine where a reasonable estimate of horizontal sensitivity could be determined and the AISA data were convolved with a sensitivity kernel before applying the retrieval. If this caused the LWP retrieval to look more like the 100+ m rows of Figs. 10 and 11, how would it affect the authors' top-level claims and interpretations of fine-scale structure?**

   We agree with the reviewer that the aspects related to 3D cloud geometry and 3D radiative transfer were not adequately addressed in the original manuscript. In the revised manuscript, we added two exercises to justify the interpretation of the AISA Hawk data. In order to account for the horizontal photon transport occurring in the considered case studies, a convolution kernel was derived by comparing 3D radiative transfer simulations performed with the Monte Carlo Atmospheric Radiative Transfer Simulator (MCARaTS, Wang et al., 2012) with 1D radiative transfer simulations performed with the Library for Radiative transfer (libRadtran) code (Mayer and Kylling, 2005; Emde et al., 2016). A narrow stripe of pure-ice was embedded in a pure liquid stratiform cloud, causing a recognizable narrow peak in the reflected cloud top radiance. The cross-correlation between the simulated 1D and 3D radiance for different

sun-sensor geometries (i.e. different solar azimuth angles and different viewing angles) was used to determine the convolution kernel. The AISA Hawk measurements were deconvolved using the Richardson-Lucy deconvolution algorithm (Richardson, 1972; Lucy, 1974), similarly to Zinner et al. (2006). The cloud top properties ($R_{1240}$, $\mathcal{I}_\mathrm{s}$, $\tau$, $r_\mathrm{eff}$, and $LWP$) were recalculated based on the deconvolved radiance fields.

Once the radiance fields are deconvolved, the observed small-scale features on the cloud top become more apparent. Regions of low reflected radiance present lower values and viceversa for areas with high radiance. For the presented case studies, this causes the relationship between $R_{1240}$ and $\mathcal{I}_\mathrm{s}$ to spread over a slightly larger range of values than originally, but it maintains its general behavior, with $R_{1240}$ and $\mathcal{I}_\mathrm{s}$ increasing simultaneously on 25 May and presenting opposite trends on 2 June. In the same way, the retrieved $LWP$ keeps its features: on 25 May it increases towards optically thicker areas, indicating the natural variability of liquid water top layer. On the contrary, on 2 June it presents anomalously high values, specially in areas of low reflectivity, which results from neglecting the ice in the retrieval of $\tau$ and $r_\mathrm{eff}$, as the reviewer as well notes in comment 2. As the deconvolution did not change the general features of $R_{1240}$, $\mathcal{I}_\mathrm{s}$, and the retrieved $LWP$, the conclusions drawn in the original manuscript still hold.

The convolution kernel applies only for horizontal photon transport, but does not account for shadows in the AISA Hawk fields. In a second exercise, we estimated the effect of the 3D cloud structure on the retrieved cloud top properties. The correction of self-shading artifacts would require highly spatially resolved information of the cloud top geometry. Although AMALi (lidar) provides the cloud top height of the nadir cloud section, its 1 s sampling rate does not allow a thorough reconstruction of the cloud top structures. Additionally, no information about the cloud regions out of nadir is available. Therefore, the correction of the radiance fields from self-shading artifacts is not possible. However, 3D radiative transfer simulations allow to interpret the effects that shading artifacts and an inhomogenous spatial distribution of the thermodynamic phase have on the cloud top properties $R_{1240}$, $\mathcal{I}_\mathrm{s}$, and $LWP$. These effects were analyzed using idealized 3D radiative transfer simulations. It has been found that, although shaded regions may present an increment in $\mathcal{I}_\mathrm{s}$ related to a lower $R_{1240}$, an associated anomalously increase in the retrieved $LWP$ is not observed in the simulations. However, simulations with an inhomogeneous distribution of cloud phase result in an increase of the retrieved $LWP$. This increase results from assuming pure liquid water clouds in the $LWP$ retrieval procedure. Therefore, the observations of 2 June can be explained by the distribution of ice and liquid water, but are not significantly effected by shades.

Changes:

- A new Appendix A, including detailed explanation of the procedure to derive the convolution kernel and results of 3D simulations to analyze the effects of shades and phase inhomogeneities, was included.

- p4 l13: *"At this resolution, horizontal photon transport needs to be taken into account. The AISA Hawk measurements have been corrected from this effect using the deconvolution algorithm introduced in App. A"* was included.

- p6 l12: *"Appendix A analyzes the different impact of shades and inhomegeneous thermodynamic phase distributions in the retrieved LWP"* was included.

- Tables 1 and 2 were updated.

- Figures 4, 5, 6, 7, 8, 9, 10, and 11 were updated.

- Figures 4 and 6 present now a narrower swath than before because artifacts resulting from the deconvolution procedure were removed. For the same reason, in the resolution analysis in Sect. 5 the pixel sizes were adapted to the new available data (instead of 25 m, 100 m, 500 m, and 1000 m pixel sizes, 30 m, 90 m, 450 m, and 900 m have been considered).

2. **Accuracy of the slope phase index for characterizing mixed phase clouds. While $\mathcal{I}_\mathrm{s}$ has been demonstrated as an effective way to discriminate pure clouds, it is not always clear how to interpret $\mathcal{I}_\mathrm{s}$ for mixed phase clouds since, as the authors note, the vertical partitioning can favor one or the other. In fact, the particles can themselves be mixed together at \*very\* small spatial scales - such intimate mixing could contribute to the strong overestimation of $LWP$ for the warm air advection scenario (as noted at the bottom of page 11). Even the interpretation as intimately-mixed particles is suspect though since the "fishnet" manifold is so tight near the AISA data in Fig. 8. Does the ratio really provide enough measurement power to discriminate mixed phases, particularly given the uncharacterized uncertainties I mention above - how can we exclude self-shading and sky view fractions as an alternative account for the apparent difference between "Holes" and "Domes?"**

We agree with the reviewer about the limited capabilities of $\mathcal{I}_\mathrm{s}$ alone to identify mixed-phase cases, as we explain in Sect. 6 (p22 l17-20). A typical photon penetration depth inferior to $200\,\mathrm{m}$ in the wavelength range defining $\mathcal{I}_\mathrm{s}$ (Ehrlich et al., 2009) prevents its sensitivity to phase inhomogeneities occurring in low cloud layers, as it is the case for 25 May. Nevertheless, in this case, the misclassification of the scene as pure liquid produces no further artifacts on the retrieved cloud properties, which also relate to the pure liquid layer top and yield a value of the retrieved $LWP$ comparable to that based on the $89\,\mathrm{GHz}$ channel of MiRAC. The correct classification of these stratified distributions of cloud phase as mixed-phase based on $\mathcal{I}_\mathrm{s}$ is only possible if information on the cloud vertical structure, e.g. provided by active remote sensing, is available.

The $\mathcal{I}_\mathrm{s}$ on 2 June presents two clear differences compared to the one on 25 May: it increases its value in areas of low $R_{1240}$ and the values cannot be reproduced by simulations of pure liquid water clouds (Fig. 7). Additionally, the $LWP$ retrieved based on the AISA Hawk radiance is on average higher than the retrieval of MiRAC and acquires specially high values in areas of low $R_{1240}$ (Fig. 6i). 3D radiative transfer simulations included in the new App. A show that, although shaded areas in the image may produce an increment in $\mathcal{I}_\mathrm{s}$, they cannot account for the high values of $LWP$ if no increase of the ice concentration is present. Therefore, it is only the combination of $R_{1240}$, $\mathcal{I}_\mathrm{s}$, and the retrieved $LWP$ that allows the identification of mixed-phase scenes. Even after combining the information provided by these three quantities, Fig. 8 highlights that the mixed-phase identification is only possible if the ice phase appears directly at cloud top.

The classification between "domes" and "holes", based on the 25th percentile of $R_{1240}$ and the 75th percentile of $\mathcal{I}_\mathrm{s}$, isolates the small-scale features observed on the cloud top geometry (the "holes" correspond to the overlayed contours of Figs. 6a and 6c). Although it is not possible to assure that the holes are not influenced by shadows, their enhanced $LWP$ with respect to the "domes" regions (Fig. 6i) supports our hypothesis of mixed-phase over shades effects, similar to the simulations in App. A. The comparison of the measurement, classified in holes and domes, with the mixed-phase scenarios presented in Figs. 8b and 8c highlights that the different $R_{1240}$ and $\mathcal{I}_\mathrm{s}$ observed in the cloud holes compared to the cloud domes can be due to several reasons. According to these two mixing-scenarios, the high values of $\mathcal{I}_\mathrm{s}$ in areas of low $R_{1240}$ are due to either larger ice fractions or larger particle sizes. However, as we mention on p16 l3-7, the large possibilities to combine $IWP$, $LWP$, $r_{\mathrm{eff,ice}}$, and $r_{\mathrm{eff,liquid}}$ as input to the radiative transfer simulations, make it impossible to find the perfect mixing scenario that fits the observations completely and to conclude which mixing scenario was more likely to happen. Using more realistic vertical profiles of $LWC$, $IWC$, $r_{\mathrm{eff,ice}}$, and $r_{\mathrm{eff,liquid}}$ provided by the LES simulations can shed light over this issue.

Changes:

- p22 l17 was updated to *"The comparison of both cloud cases highlights the limitations of passive remote sensing alone to identify layered mixed-phase structures if the ice is not sufficiently close to the*

*cloud top. In particular in these cases, the combination of active and passive remote sensing is crucial to fully characterize the horizontal and vertical distribution of ice and liquid particles in mixed-phase clouds."*

3. **Formal hypothesis testing, or uncertainty quantification or propagation. The general character of the manuscript is to bring together multiple measurement modes ($LWP$ by AISA and MiRAC) and conclude that both are important to interpret diverse cloud structure. Fair enough, but do the authors have an uncertainty budget for either instrument? For example, what is the error in the Backscatter - the vertical profiles seem very qualitative in nature and it is unclear to what degree the temporal axis represents meaningful change in structure. This is important in comparing said structure to the $LWP$ maps - which are themselves uncertain up to a level determined by the instrument noise and unknowns in the observation system. LES analysis is of a similarly qualitative nature, and an important first step which is meritorious as an exercise, but given the large mismatch between distributions of simulated and measured $\mathcal{I}_s$ it is not clear that the LES has successfully explained or even accounts for the observations. Can the authors formulate this as a formal hypothesis test of some kind?**

    (a) *Uncertainty*

    The uncertainty of the AISA Hawk radiance fields is a combination of the calibration uncertainty and noise in the measured signal and has been estimated to be in the range of 6 % (Schäfer et al., 2013). Error propagation allows the estimation of the uncertainty in the cloud top properties and eventually yields a $LWP$ between $2.0\,\mathrm{g\,m^{-2}}$ and $36\,\mathrm{g\,m^{-2}}$ with a 36 % uncertainty for case on 25 May and a $LWP$ between $166\,\mathrm{g\,m^{-2}}$ and $380\,\mathrm{g\,m^{-2}}$ with a 32 % uncertainty for the case on 2 June.

    The uncertainty of the $LWP$ retrieval based on measurements of the 89 GHz passive channel of MiRAC is addressed in App. B (former App. A). The major source of uncertainty results from the determination of the clear sky brightness temperature, TB0. The two considered cases have an estimated uncertainty in the range of 5 % to 6 % ($5\,\mathrm{g\,m^{-2}}$ for $100\,\mathrm{g\,m^{-2}}$ $LWP$ and $12\,\mathrm{g\,m^{-2}}$ for $200\,\mathrm{g\,m^{-2}}$ $LWP$). Therefore, the MiRAC $LWP$ on 25 May ranges between $20\pm1\,\mathrm{g\,m^{-2}}$ and $40\pm2\,\mathrm{g\,m^{-2}}$, in agreement with the retrieval of AISA Hawk for the thicker cloud regions. On 2 June, the MiRAC $LWP$ ranges between $90\pm5\,\mathrm{g\,m^{-2}}$ and $120\pm7\,\mathrm{g\,m^{-2}}$, presenting lower values than the AISA Hawk $LWP$.

    The radar reflectivity of MiRAC, shown in Fig. 3, has an uncertainty of 0.5 dB (Mech et al., 2019). The lidar backscatter of AMALi is defined as in Langenbach et al. (2019). Its uncertainty is calculated using a maximum-error propagation which takes into account signal-inherent error sources, like noise, and uncertainties from assumptions in the retrieval. However, in this study, the lidar backscatter signal is interpreted only qualitatively to determine the cloud top altitude and the upper-most cloud structure relative to increments in the radar signal, hinting the existence and thickness of the liquid top layer. The geometry of the backscatter signal was corrected using the aircraft GPS and INS data. The magnitude of the lidar backscatter signal is used only to differentiate clean air from cloudy regions, with Fig. 3 displaying only the latest. A backscatter ratio of 20 (i.e. $\log(\mathrm{BSC532})=3$) is 20 times larger than the signal of clear air and was and was chosen as threshold to discriminate the cloud signal from any (much lower) aerosol signal (Gutleben et al., 2019).

    Changes:

    - p4 l11: *"Considering uncertainties due to the calibration procedure and noise in the measured signal, the uncertainty in the radiance is estimated to be in the range of 6 % (Schäfer et al., 2013)"* was included.

- p8 l4: *"The combination of measurements is interpreted qualitatively to gain an insight into the clouds vertical structure"* was included.

- p8 l5: *"defined as in Langenbach et al. (2019)"* was included.

- p8 l25 was updated to *"Mean values and associated uncertainty of the cloud properties are summarized in Tab. 1"*.

- p11 l4 was updated to *"Table 1 presents the mean values and associated uncertainty the presented cloud properties"*.

- The uncertainty of the cloud top properties was included in Tab. 1 and its caption was updated to *"Average value and uncertainty ($\Delta$) of the cloud top properties derived from the measurements of AISA Hawk on 25 May and on 2 June. Independent estimations of the LWP range by the passive 89 GHz channel of MiRAC are also included."*

(b) *Large Eddy Simulations*

It is true that ICON-LEM did not fully reproduce the observations. ICON-LEM is a relatively new model and its performance in Arctic conditions is not yet completely evaluated (Schemann and Ebell, 2020). However, based on the comparisons with the observations, we tried to identify which aspects of the model did lead to the observed discrepancies, e.g. low ice crystal concentration. Therefore, for 25 May, the ICON-LEM produced $LWC$ and $IWC$ profiles (Fig. 9a) are in general in agreement with the active remote sensing in Fig. 3a (i.e. mainly liquid at cloud top from which ice precipitates). The associated $R_{1240}$ and $\mathcal{I}_s$ reproduce the AISA Hawk measurements on that day, being mainly affected by the higher concentration of liquid water at cloud top and presenting typical values corresponding to liquid water clouds. The knowledge of the cloud vertical structure provided by ICON-LEM allows manipulating the fields with the purpose of understanding how the cloud $R_{1240}$ and $\mathcal{I}_s$ react under different theoretical scenarios. In this way, an additional test case where the ICON-LEM $IWC$ was doubled was included. Even with this anomalously increased ice concentration, $R_{1240}$ and $\mathcal{I}_s$ keep being influenced mainly by the liquid top layer.

On 2 June, the original ICON-LEM $IWC$ is negligible. Hence, the associated $R_{1240}$ and $\mathcal{I}_s$ agree with simulations corresponding to liquid water clouds, but do not reproduce the AISA Hawk measurements on that day. Artificially enhancing 1000 times the $IWC$ provided by ICON-LEM in order to obtain values in the same order of magnitude than the $LWC$, reveals its proximity to the cloud top and originates an observable impact on $R_{1240}$ and $\mathcal{I}_s$. Their values spread towards the pure ice region and now reproduce the AISA Hawk observations. The reasons why ICON-LEM appears to produce so little ice on this day are unclear. Nevertheless, combining LES with radiative transfer simulations and its comparison to spectral measurements has the potential to identify misperformances of ICON-LEM, which can be used to improve its internal parameterizations.

When comparing ICON-LEM and observations, the setup of this comparison needs to be considered carefully. The larger spread of the ICON-LEM data in the $R_{1240}$ - $\mathcal{I}_s$ diagram compared to the measurements of AISA Hawk is due to the larger spatial area considered in the LES (Fig. 2). In this larger region, the LES cloud field has a higher variability than the limited narrow line of the measurements, e.g. the ICON-LEM domain features cloud holes which are not overflown by AISA Hawk. We selected this larger domain of ICON-LEM in order to obtain a statistically robust number of grid points.

Changes:

- The description of ICON-LEM and its setup was moved to the new subsection *2.3 Large Eddy Simulations (LES)*. At the beginning, we estate: *"Simulations using the ICOsahedral Non-hydrostatic atmosphere model (ICON), operated in its Large Eddy Model (LEM) configuration (Heinze et al.,*

*2017; Dipankar et al., 2015) provide a quantitative view into the cloud vertical structure. The simulated cloud vertical profiles were used as input for radiative transfer simulations to analyze the impact of different vertical distributions of the cloud phase on the cloud top horizontal variability".*

- Section 4 was shortened and renamed to *Comparison of measurements and LES*. Its beginning was updated to *"Comparing simulated cloud reflectivities and phase index based on ICON-LEM cloud fields with the measurements of AISA Hawk will help to evaluate the conclusions about the vertical structure of the cloud thermodynamic phase drawn in the previous section"*.

- p18 l3: *"located outside the measurement area of AISA Hawk"* was included.

- p18 l18: *"which were not observed by AISA Hawk"* was changed to *"which were located outside the AISA Hawk measurement region"*.

**Minor technical clarification:**

**How do the horizontal (i.e.) spatial scales compare between Fig. 3 and the AISA data?**

The temporal axis of Figs. 3a and 3b coincides with the temporal axis of Figs. 4 and 6, respectively. Therefore, the measurements in Figs 3a and 3b cover the same distance in flight direction than Figs. 4 and 6 (i.e. 4.7 km on 25 May and 5.6 km on 2 June). With a sampling frequency of 1 s and 1.3 s respectively, AMALi and MiRAC have a spatial resolution of 80 m and $\sim$100 m on 25 May (80 m s$^{-1}$ flight speed) and 90 m and $\sim$120 m on 2 June (90 m s$^{-1}$ flight speed).

Changes:

- Spatial axes were included in Fig. 3.

**Answers to Anonymous Referee #2**

**General comments**

**This paper uses a combination of active and passive remote sensing instrumentation to characterize the small scale structure of cloud thermodynamic phase using two case studies observed in the Arctic. The paper is well written, gives significant results and the approach seems technically sound. Thus, as such, I think it is appropriate for publication in ACP. However, there are a few minor editorial comments and some additional points that the authors might want to consider before the publication is finalized.**

We thank the reviewer for her/his careful reading of the manuscript and her/his suggestions. With our reply and the revised version of the manuscript, we hope to address all the comments in a satisfying way. Our answers are structured and indicated as follows: reviewer comments (bold), answers, changes in the manuscript (italic).

**I would like to see more explanation on why the two particular case studies were chosen and how representative these case studies are of conditions encountered in the Arctic in general. Although cases of single-layer mixed-phase clouds do occur in the Arctic as the authors state, and although they are nice to examine from a process oriented perspective because it involves the complications of interactions between different cloud layers, past studies have suggested that multi-layer clouds and even multilayer mixed-phase clouds may be more common than these single-layer clouds. Thus, some explanation of how the results from these special cases are applicable in general to remote sensing (especially cases when active remote sensing data are not available) would be appropriate.**

The two cloud cases were chosen because they represent two contrasting cases of Arctic single-layer boundary-layer clouds (i.e., warm air advection vs. cold air outbreak), providing different conditions for cloud remote sensing. Even when categorized into warm air advection and cold air outbreak, we do not intend to draw general conclusions about the cloud properties of these scenarios. This could only be done on statistical basis, analyzing more cloud cases.

As we showed in the manuscript, the interpretation of passive remote sensing is already challenging for single-layer mixed-phase clouds. Therefore, the study focused on this specific type of Arctic clouds and excluded cases of multi-layer clouds. However, boundary-layer clouds are very common in the Arctic and, even in the case of multiple-layer clouds, they often contain a liquid-dominated top layer, similar to the two case studies (Mioche et al., 2015). Unfortunately, high cloud layers extending beyond the boundary layer cannot be analyzed with the unpressurized Polar 5 aircraft.

Changes:

- p2 l9: *"Cold air outbreaks occur all year long but they are especially frequent in winter (Kolstad et al., 2009; Fletcher et al., 2016). Warm and moist air masses intruding the Arctic from southern latitudes occur 10 % of the time all year long and are responsible for most of the transport of moisture and heat into the Arctic (Woods et al., 2013; Sedlar and Tjernström, 2017; Pithan et al., 2018)"* was included.

- p7 l26: *"Cold air outbreaks and warm air advections are phenomena often affecting the Arctic regions (Pithan et al., 2018; Sedlar and Tjernström, 2017; Woods et al., 2013; Kolstad et al., 2009; Fletcher et al., 2016). The occurrence of both situations during the ACLOUD campaign make it an ideal testbed to contrast the characteristics of the clouds occurring under each situation."* was included.

- p22 l17 was updated to *"The comparison of both cloud cases highlights the limitations of passive remote sensing alone to identify layered mixed-phase structures if the ice is not sufficiently close to the cloud top. Specially in these cases, the combination of active and passive remote sensing is crucial to fully characterize the horizontal and vertical distribution of ice and liquid particles in mixed-phase clouds."*

**Were there any Doppler radar data available? Some past studies have shown that the presence of cloud top generating cells frequently occur in the Arctic (as well as in other regions) and could be responsible for some of the horizontal inhomogeneity. If such data are available, perhaps more could be said about the scales of mixing of the phases and their horizontal distributions (and the processes). This would also give more information about the resolution required for analysis.**

Unfortunately, no Doppler radar data are available. In order to detect the low-level Arctic clouds occurring only a few hundred meters above the ground, MiRAC was mounted on Polar 5 with a 25° angle. This increases the vertical resolution and decreases the impact of the ground return, thus reducing the blind zone above the ground (Mech et al., 2019). However, it causes as well a very high horizontal wind component dragging the cloud particles along. This prevents for a reliable estimate of the particle vertical velocity.

**My other concern relates to the use of hexagonal columns to characterize the ice crystals. A lot of previous studies have suggested that the majority of ice crystals in Arctic clouds, including those in mixed-phase conditions, are very irregular and not well characterized by pristine shapes. Can a more realistic assumption about the ice crystal shapes be used? Or, alternatively, there should be more discussion made about the quantitative uncertainties induced by this simplistic assumption.**

It is true that irregular ice crystals are more likely in Arctic mixed-phase clouds. However, from a radiative transfer perspective, we interpret the spectral absorption features of ice crystals. As shown by Ehrlich et al. (2008a,b), the cloud top reflectivity features occurring in the spectral range considered in this study are mainly dominated by the absorption features of the ice, determined in turn by the imaginary part of the refractive index, and the ice crystals effective radius, $r_{\text{eff}}$. We tested different non-spherical ice crystal shapes in the radiative transfer simulations presented here and obtained similar results. Accounting for hexagonal columnar ice crystals introduces a negligible bias compared to considering a mixture of ice crystal habits, and does not change the conclusions drawn from the qualitative comparison between measurements and simulations.

Changes:

- p5 l18: *"Regarding the phase index, Ehrlich et al. (2008a,b) found that the influence of the ice crystal shape is of minor importance compared to the impact of the particle size. Additional simulations considering different ice crystal habits (not shown here) confirmed this. Hence, the assumption of columns is sufficient to account for the non-sphericity effects of the ice crystals."* was included.

**Page 4, line 30: It is not true in general that the radar reflectivity for ice is proportional to the sixth power of the ice particle size. For example, Hogan and collaborators have developed much better quantitative models for converting ice crystal particle size distributions to radar reflectivity.**

We are sorry for this unprecise phrasing. We meant that the radar reflectivity is proportional to the sixth moment of the particle size distribution (Hogan and O'Conner, 2004) and thus, it is more sensitive to large particles, which are likely to be ice crystals. Since lidar signals are more sensitive to smaller and populous particles like liquid water droplets, Shupe (2007) and Kalesse et al. (2016) point towards a synergetic use of lidar and radar measurements to identify different in-cloud phase regimes similar to what presented here.

Changes:

- p5 l3 was updated to *" The radar reflectivity is proportional to the sixth power of the particle size distribution, and thus, is most sensitive to large particles, such as ice crystals (Hogan and O'Conner, 2004; Shupe, 2007; Kalesse et al., 2016). Therefore, it is used as an indicator of the vertical location of large ice crystals in mixed-phase clouds."*

**Page 5, line 8: How can the standard profile be used in combination with the dropsonde data? Wouldn't one or the other have to be used to give the vertical profile?**

The dropsonde data does not provide all the parameters required for the radiative transfer simulations. By combining the dropsonde data with the standard atmospheric profiles, libRadtran makes use of the measured pressure, temperature, and humidity profiles and scales the gas species concentrations ($O_3$, $NO_2$, BrO, $ClO_2$, $CH_2O$, $SO_2$, and $O_4$) of the standard atmospheric profile to the measured pressure.

**Page 6, line 3: What are unrealistic values of LWP? This should be more quantitative.**

Shupe et al. (2006), Mioche et al. (2017), Nomokonova et al. (2019) and Gierens et al. (2019) report values of the $LWP$ rarely exceeding $100\,\mathrm{g\,m^{-2}}$, with mean values between 30 and $50\,\mathrm{g\,m^{-2}}$. Therefore, we conclude that the retrieved values based on the measurements of AISA Hawk (in average $270\,\mathrm{g\,m^{-2}}$) are caused by a retrieval bias due to the existence of ice in the considered clouds.

Changes:

- p6 l10: *"Past observations show that the $LWP$ of typical Arctic boundary-layer clouds is on average 30 - $50\,g\,m^{-2}$ and rarely exceeds $100\,g\,m^{-2}$ (Shupe et al., 2006; Mioche et al., 2017; Nomokonova et al., 2019; Gierens et al., 2019)"* was included.

**Page 6, line 6: What quantitative criteria were used to identify the presence of cold air outbreaks?**

The synoptic conditions during ACLOUD were analyzed by Knudsen et al. (2018). Within their study, the Marine Cold Air Outbreak (MCAO) index is analyzed. Similar to Kolstad (2017) and Papritz et al. (2015), the MCAO index is based on the difference between the surface and the $850\,\mathrm{hPa}$ potential temperature, which was calculated every 6 hours during the ACLOUD period. Whereas possitive values of the MCAO index were used to identify cold air outbreak events, strong negative values of the MCAO index with respect to the climatology identified warm air advection events. Based on this analysis, Knudsen et al. (2018) divided the ACLOUD period into a cold period (between 23 and 29 May 2017, characterized by positive values of the MCAO index), a warm period (between 30 May and 12 June 2017, with values of the MCAO index well below the climatology) and a normal period (between 13 and 26 June 2017, where the MCAO index acquired similar values to the climatology).

**Page 7, line 6: Why couldn't it also be attributed to a reduced concentration of particles rather than just small particles?**

This is correct, a reduced concentration of particles would lead to a similar effect. Thanks for pointing on this insufficient explanation.

Changes:

- p8 l18 was updated to *"either to smaller ice crystals or to a reduced particle concentration"*.

**Additional changes:**

[revised manuscript text omitted]